# Active self-assembly of piezoelectric biomolecular films via synergistic nanoconfinement and in-situ poling

Zhuomin Zhang [1,2,9], Xuemu Li[1,2,9], Zehua Peng[1,2], Xiaodong Yan[1,2], Shiyuan Liu[1,2], Ying Hong [1,2], Yao Shan[1,2], Xiaote Xu[1,2], Lihan Jin[2], Bingren Liu[2], Xinyu Zhang[3], Yu Chai[3], Shujun Zhang [4] ✉, Alex K.-Y. Jen [5,6,7,8] ✉ & Zhengbao Yang [1,2,5] ✉

Piezoelectric biomaterials have attracted great attention owing to the recent recognition of the impact of piezoelectricity on biological systems and their potential applications in implantable sensors, actuators, and energy harvesters. However, their practical use is hindered by the weak piezoelectric effect caused by the random polarization of biomaterials and the challenges of large-scale alignment of domains. Here, we present an active self-assembly strategy to tailor piezoelectric biomaterial thin films. The nanoconfinement-induced homogeneous nucleation overcomes the interfacial dependency and allows the electric field applied in-situ to align crystal grains across the entire film. The β-glycine films exhibit an enhanced piezoelectric strain coefficient of 11.2 pm V$^{-1}$ and an exceptional piezoelectric voltage coefficient of $252 \times 10^{-3}$ Vm N$^{-1}$. Of particular significance is that the nanoconfinement effect greatly improves the thermostability before melting (192 °C). This finding offers a generally applicable strategy for constructing high-performance large-sized piezoelectric bio-organic materials for biological and medical microdevices.

In 1880, Curie brothers first discovered and demonstrated the piezoelectric effect, which is an intrinsic property of crystals with a non-centrosymmetric structure that allows robust and precise conversion between electrical and mechanical energies. The extensive and ongoing research on advanced piezoelectric materials has benefited a broad range of applications in actuators, sensors, acoustic transducers, energy harvesters, wastewater treatment, and catalysis[1–6]. Although people are endeavoring to develop synthetic piezoelectric materials, nature seems to have grasped the effect for millions of years[7–11]. Piezoelectric biomaterials have superiority over the widely used piezoceramics for biotechnology applications since they naturally exhibit biocompatibility, accessibility, and environmental sustainability[12,13]. However, most research on piezoelectric biomaterials remains in theoretical analysis. The challenges in aligning the domain orientation and the weak piezoelectric effect greatly limited their applications.

Although various self-assembly or template-assisted assembly methods have been developed to synthesize piezoelectric biomaterials, most techniques are relatively complex to scale up and challenging to achieve the strongest polar orientation in the out-of-plane (OOP) direction[14–20]. Their polarization direction is either antiparallel

[1]Department of Mechanical and Aerospace Engineering, Hong Kong University of Science and Technology, Clear Water Bay, Hong Kong, China. [2]Department of Mechanical Engineering, City University of Hong Kong, Hong Kong, China. [3]Department of Physics, City University of Hong Kong, Hong Kong, China. [4]Institute for Superconducting and Electronic Materials, Australian Institute of Innovative Materials, University of Wollongong, Wollongong, NSW, Australia. [5]Department of Materials Science and Engineering, City University of Hong Kong, Kowloon, Hong Kong. [6]Department of Chemistry, City University of Hong Kong, Kowloon, Hong Kong. [7]Hong Kong Institute for Clean Energy, City University of Hong Kong, Kowloon, Hong Kong. [8]Department of Materials Science and Engineering, University of Washington, Seattle, WA 98195-2120, USA. [9]These authors contributed equally: Zhuomin Zhang, Xuemu Li. ✉e-mail: shujun@uow.edu.au; alexjen@cityu.edu.hk; zbyang@ust.hk

in-plane (IP) or at a certain angle with the out-of-plane direction, which greatly weakens their piezoelectricity. Among the biomaterials, glycine, the simplest non-chiral amino acid, has three distinct crystallization polymorphs, non-piezoelectric α-glycine, piezoelectric β-glycine, and piezoelectric γ-glycine. β-glycine crystals exhibit high shear piezoelectricity (178 pm V[−1]) and marvelous piezoelectric voltage coefficient (8 Vm N[−1]) larger than any currently used ceramic or polymer[21]. Unfortunately, β-glycine is the most difficult to form in kinetics and the most unstable in thermodynamics under ambient conditions[22]. The excessively high coercive electric field also makes it quite challenging to polarize glycine crystals and align the domains at the macroscale, even though they are ferroelectric[23].

Here, we report a strategy to fabricate piezoelectric β-glycine films resembling the inorganic polycrystalline morphology. This is inspired by the century-long research on piezoceramics, represented by lead zirconate titanate (PZT). Piezoceramics have always been taking the dominant role thanks to their properties of tunable piezoelectricity, excellent stability, low cost, and easy preparation[24]. They can be constructed in desired shapes and diverse sizes and evolved into piezoelectric metamaterials or flexible composites[25–28]. Hence, we deliberately fuse the nanoconfinement and in-situ electric fields to the nucleation and self-assembly of the biomolecular glycine, mimicking the sintering and poling processes in piezoceramic manufacturing. During the synthesis, the electric field is used not only to generate the nanocrystals of β-glycine but also plays the role of in-situ poling that facilitates the domain alignment across the entire film. The dense and continuous films exhibit outstanding piezoelectricity, property uniformity, as well as anomalously excellent thermodynamic stability resulting from the nanoconfinement effect.

## Results

### Piezoelectric biomolecular films fabrication and active self-assembly mechanism

The β-glycine nanocrystalline films are fabricated based on a bio-organic film printer using the electrohydrodynamic spray method (Fig. 1a, Supplementary Note 1, and detailed processes are shown in the Methods section). During the spray process, an electric field is applied between the nozzle tip and the conductive support to overcome the surface tension of the glycine aqueous solution, producing numerous nano-micro droplets (Fig. 1b, Supplementary Fig. 1, and Supplementary Movie 1)[29,30]. With the rapid water evaporation and the increasingly large surface-area-to-volume ratio of the nano-micro droplets, the glycine nucleus is formed in the β phase through the nanoconfinement effect. Whereas the α polymorph forms most readily when glycine is crystallized from aqueous solutions, the metastable β polymorph has been demonstrated to be favored in nanoscopic pores or micrometer-scale patterned substrates[31,32]. This is supported by the Ostwald step rule that the least stable polymorph crystallizes first at the early stages of the crystallization because of its small size. It can be explained using the classical nucleation theory[33]. The free energy of the nucleus is equal to the sum of the volume free energy change $\Delta G_V$ and adverse surface free energy $\Delta G_S$ (Fig. 1c). With regard to the typical spherical nucleus, the free energy along the crystallization path can be described as follows:

$$\Delta G_{cryst} = \Delta G_V + \Delta G_S = \frac{4}{3}\pi r^3 \Delta g + 4\pi r^2 \sigma \tag{1}$$

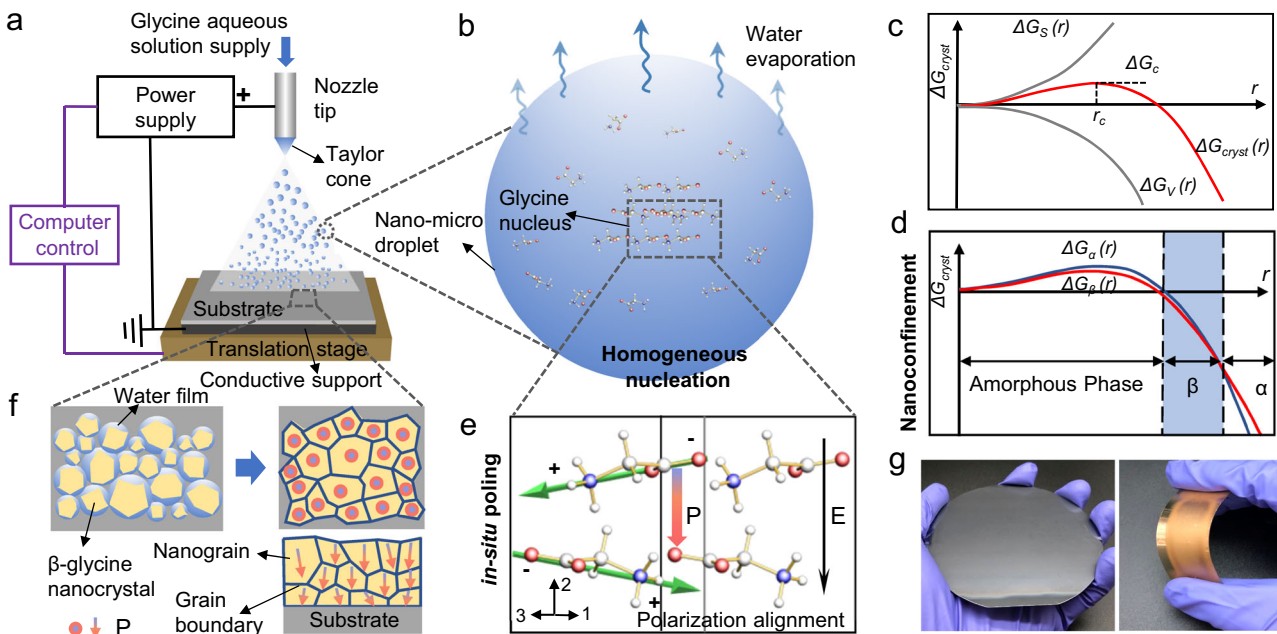

**Fig. 1 | Fabrication of piezoelectric β-glycine nanocrystalline films and the active self-assembly mechanism via synergistic nanoconfinement and in-situ poling. a** Schematic of the bio-organic films printer and the synthesis of β-glycine nanocrystalline films. **b** Schematic of the nano-micro droplet of glycine solution and the crystallization process. **c** Illustration of the free energy ($\Delta G_{cryst}$) profile of a growing crystal nucleus as a function of crystal radius, *r*. The energy profile results from the sum of the favorable volume free energy, $\Delta G_V$, and the surface free energy $\Delta G_S$. The profile passes through a maximum value of $\Delta G_{cryst}$ at the critical radius, $r_c$. **d** Illustration of the size-dependent free energy profiles for two competing nuclei corresponding to α-glycine and β-glycine. **e** Schematic of orientation alignment of glycine molecules during homogeneous nucleation. Molecular dipoles in β-glycine sum to a spontaneous polarization (red arrow P) along the 2-axis parallel to the electric field (black arrow E), which contributes to the longitudinal 22 piezoelectric coefficient. Molecules are displayed in the CPK coloring, including carbon (cyan), hydrogen (white), oxygen (red), and nitrogen (navy blue) atoms[48]. The green arrow represents the dipole orientation of individual glycine molecule. **f** Schematic of the film formation process showing the compact nanograins with uniform and consistent polarization orientation (red spot and red arrow in nanograins). The top two images are the surface view of the films, and the bottom image is the cross-sectional view. **g** Photographs of a film on a 4-inch silicon wafer (left) and film on a flexible gold-coated polyethylene terephthalate (PET) substrate (right).

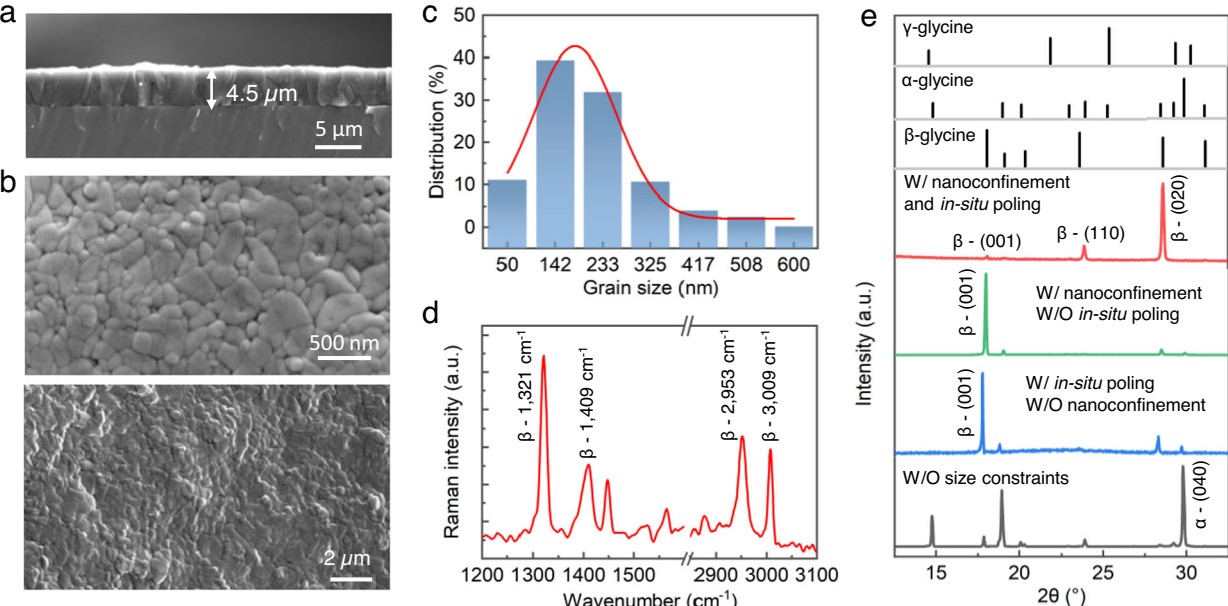

**Fig. 2 | Morphology and structural characterization of β-glycine nanocrystalline films. a** Cross-sectional SEM image of an as-obtained film with a thickness of 4.5 μm. **b** Surface topography SEM image showing the compact nanosized grains of the uniform and continuous films. **c** Grain size distribution of β-glycine nanograins by analyzing over 400 grains. **d** Raman spectrum of the as-grown β-glycine nanocrystalline films. **e** XRD spectra of the as-prepared β-glycine nanocrystalline films (red curve), β-glycine nanocrystals obtained in the absence of electric field (green curve), β-glycine microcrystals prepared by electrohydrodynamic focusing deposition (blue curve), and α crystals formed by direct evaporation of glycine solution film (black curve). The standard XRD spectra of three phases of glycine are shown at the top of the figure. W/denotes with, and W/O stands for without.

in which $r$ is the radius of the spherical nucleus, $\Delta g$ denotes the free energy gap between the nucleated phase and the nucleating phase for a unit volume, and $\sigma$ is the surface tension of the interface and represents the surface free energy in each unit area. From Eq. (1), it is obvious that $\Delta G_{cryst}$ is strongly related to the crystal size. The maximum value of $\Delta G_{cryst}$ can be obtained by derivating it with regard to $r$, corresponding to the activation energy of nucleation $\Delta G_c$ at the critical radius $r_c$. It is critical to surmount the energy barrier for spontaneous nucleation. Because of the distinct crystal structures of polymorphs, their specific surface energies, volume free energies, and crystal morphologies should also be different. It can be reasonably inferred that each polymorph shall possess different values of $\Delta G_{cryst}$ and $r_c$. Here is where thermodynamics and kinetics collide. At the critical size, the difference in kinetic barriers of the two polymorphs is equivalent to the difference in their thermodynamic stability. When considering the trajectory of the nucleation of different forms, it is expected to have different critical sizes and different corresponding nucleation barriers. Figure 1d shows free energy profiles where α phase glycine crystals are more stable in bulk sizes, whereas metastable β phase glycine crystals are more stable at the critical size and slightly beyond. With dimensions slightly beyond the critical size, the β phase is the thermodynamically preferred phase corresponding to the lowest free energy and lower kinetic barrier compared with the other phases.

We then propose the potential nucleation and crystallization process as follows. The β-glycine nanocrystals are formed through homogeneous nucleation owing to the small size and substrate-free property of in-flight nano-micro droplets[34,35]. As homogeneous nucleation is not influenced by solid-liquid interfaces, it is possible to manipulate the crystallization process by applying external electric fields, which also serve as the poling process[35,36]. The in-situ electric field in the crystal growth process induces the domain alignment of β-glycine nanocrystals, suggesting the net polarization direction [020] is parallel to the electric field (Fig. 1e). The partially wet particles with incomplete evaporation deposited on the substrate are essential for further dense film formation in electrospray deposition

techniques[37–39]. During the synthesis of the β-glycine nanocrystalline films, the crystallization is nearly completed before depositing on the substrate, while numerous nanocrystals still carry a thin water shell and further cluster together to form compact films (Fig. 1f). Notably, the dominant OOP orientation of the nanograins remains in the strongest polar direction [020]. The as-synthesized film resembles the inorganic polycrystalline morphology, and it can be constructed in variable sizes and customizable structures on various rigid or flexible substrates (Fig. 1g and Supplementary Fig. 2).

## Morphology and structural characterizations demonstrating the role of synergistic nanoconfinement and in-situ poling on crystal grains alignment

The cross-section scanning electron microscopy (SEM) images and surface topography SEM images show the uniform and compact nanograin distribution of the β-glycine films with thicknesses ranging from 0.6 to 9 μm with the depositing speed of $1.5 \times 10^7$ μm$^3$ s$^{-1}$ (Fig. 2a, Fig. 2b, Supplementary Fig. 3, Supplementary Fig. 4, and Methods section), where the average grain size is approximately 200 nm (Fig. 2c). When in-situ heating is introduced during synthesis, the obtained films are no longer compact but rather hollow and loose because nano-micro droplets evaporate rapidly and crystallize completely without a water film before being deposited on the substrate (Supplementary Fig. 5). Raman spectra obtained from the as-prepared compact nanocrystalline films show distinct Raman shifts of β phase glycine crystals and exhibit no Raman shifts from other glycine crystalline phases, confirming that the films are almost entirely composed of piezoelectric β-glycine crystals (Fig. 2d)[40]. X-ray diffraction (XRD) spectra further confirm the pure β crystalline phase of the nanocrystalline films, where the strongest peak (020) manifests that the OOP orientation is dominated by being along the optimal piezoelectric direction (red curve in Fig. 2e).

To further demonstrate the essential role of synergizing the nanoconfinement and in-situ poling, we examine the crystallization results under three other conditions. First, we cancel in-situ electric

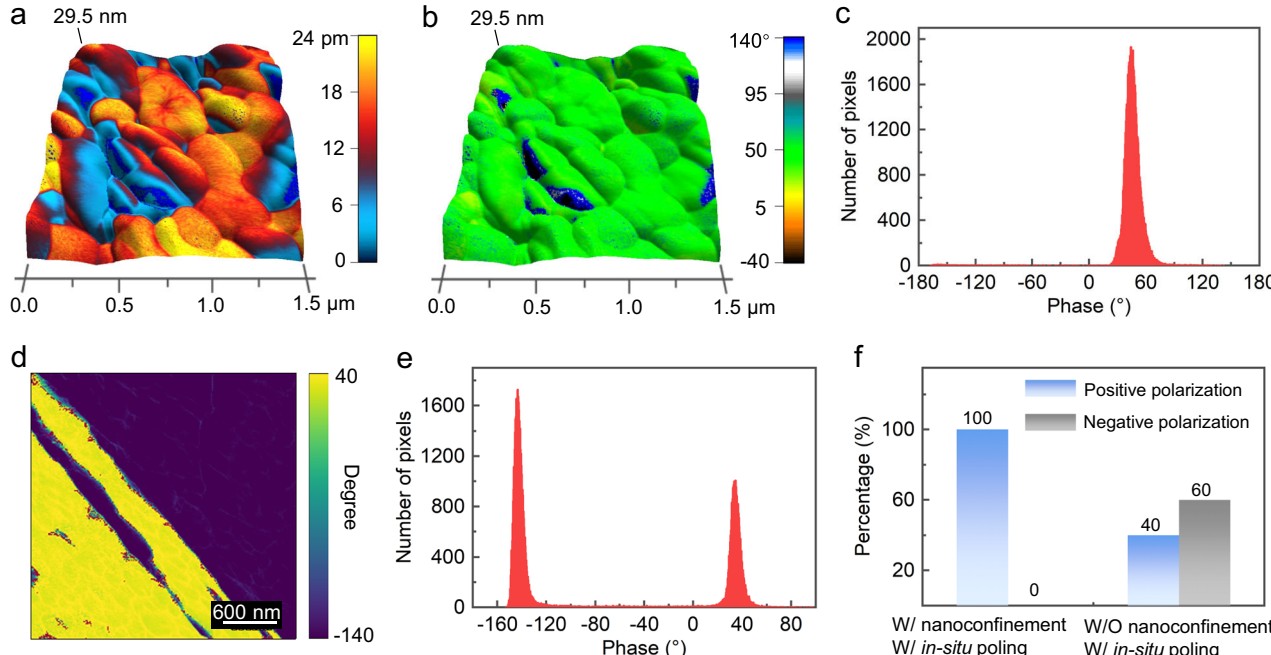

**Fig. 3 | PFM measurements and polarization alignment studies of β-glycine nanocrystalline films. a** The PFM OOP amplitude mapping overlaid on the 3D topography of as-prepared films in a 1.5 × 1.5 μm² area. The applied AC voltage is 2 V. **b** The corresponding PFM OOP phase mapping overlaid on the 3D topography. **c** Histogram calculated from the PFM OOP phase mapping in (**b**) showing that the β-glycine nanocrystalline films are dominated by domains with the unique polarization direction. **d** PFM OOP phase mapping of the β-glycine microcrystals obtained by electrohydrodynamic focusing deposition through heterogeneous nucleation. **e** Histogram calculated from the phase mapping in (**d**). **f** Comparison of statistics of the piezoelectric phase for the as-prepared β-glycine nanocrystalline films via synergistic nanoconfinement and in-situ poling (left), and β-glycine microcrystals grown by heterogeneous nucleation in the absence of nanoconfinement effect (right).

fields but keep the nanoconfinement effect by dispersing the droplets using ultrasound. In the absence of an in-situ electric field to adjust the polarization orientation, the β-glycine crystals show a dominant OOP orientation [001] that is perpendicular to the strongest piezoelectric direction [020] (green curve in Fig. 2e). Second, without introducing the nanoconfinement but still in the β phase nucleation region, i.e., using the electrohydrodynamic focusing deposition (Supplementary Fig. 6) instead of electrohydrodynamic spray, the β-glycine is crystallized into microcrystals, which are no longer formed through homogeneous nucleation but through heterogeneous nucleation on the substrate due to the interfacial effect (Supplementary Fig. 7). In this case, it is challenging to align the polarization of β-glycine microcrystals, although the external electric field is applied. The main peak of XRD spectra also shows the non-piezoelectric OOP orientation [001] (blue curve in Fig. 2e). In the final case, only α-glycine crystals are obtained by direct evaporation of glycine solution film even under a high electric field because the α phase is the easiest to form dynamically under ambient conditions without size confinement (black curve in Fig. 2e).

## PFM measurements and polarization alignment studies

The piezoelectric properties of the as-prepared β-glycine nanocrystalline films are evaluated by piezoresponse force microscopy (PFM) measurements. With the dual AC resonance tracking (DART) technique, the piezoelectric vibration induced by high-frequency drive AC voltages on the β-glycine nanocrystalline film can be examined to quantitatively determine the effective piezoelectric coefficients[41]. To ensure that the response is piezoelectric, resonance measurements are performed under different voltages (Supplementary Fig. 8). By measuring the resonance frequency and the quality factor that describes energy losses in the damped harmonic oscillator system (Supplementary Fig. 9), the intrinsic piezoresponse can finally be derived by

correcting the resonance amplification with the quality factor. Figure 3a illustrates the PFM mapping of OOP amplitudes overlaid on the 3D topography of the β-glycine nanocrystalline film, exhibiting superb and uniform piezoelectric response of the compact nanocrystals. The obvious PFM IP amplitudes and two distinct IP phases further confirm the intrinsic piezoelectricity of the films (Supplementary Fig. 10). The PFM OOP phase mapping is uniform and exhibits hardly any opposite phase, indicating that the polarization of the as-prepared films is well aligned, and the polarization direction of the nanograins is consistent (Fig. 3b). The phase histograms of Fig. 3b and the large-area phase mappings also show that the films are dominated by domains with unique polarization direction (Fig. 3c and Supplementary Fig. 11). The phase results of randomly selected regions from different samples also exhibit a uniform and consistent value, indicating that the polarization of the entire film is in the direction of the applied electric field (Supplementary Fig. 12 and Fig. 3f). When glycine is crystallized in the absence of nanoconfinement via the electrohydrodynamic focusing deposition, the PFM OOP phase mapping of the β-glycine microcrystals exhibits both domains with opposite polarizations (Fig. 3d and Supplementary Fig. 13). The phase histogram shows that the β-glycine microcrystals have an approximately equal number of domains that are 180° out of phase (Fig. 3e and Fig. 3f). Without the nanoconfinement, the β-glycine microcrystals hardly exhibit piezoelectricity at a macroscopic level since the piezoelectric effects of the opposite domains will cancel out each other, which evidences the indispensable effect of the synergy of the nanoconfinement effect and in-situ poling on the domain alignment.

## Measurements of the piezoelectric coefficients and ferroelectric responses

To further quantify the piezoelectric strength of the β-glycine nanocrystalline films, we measure the PFM amplitudes averaged over the

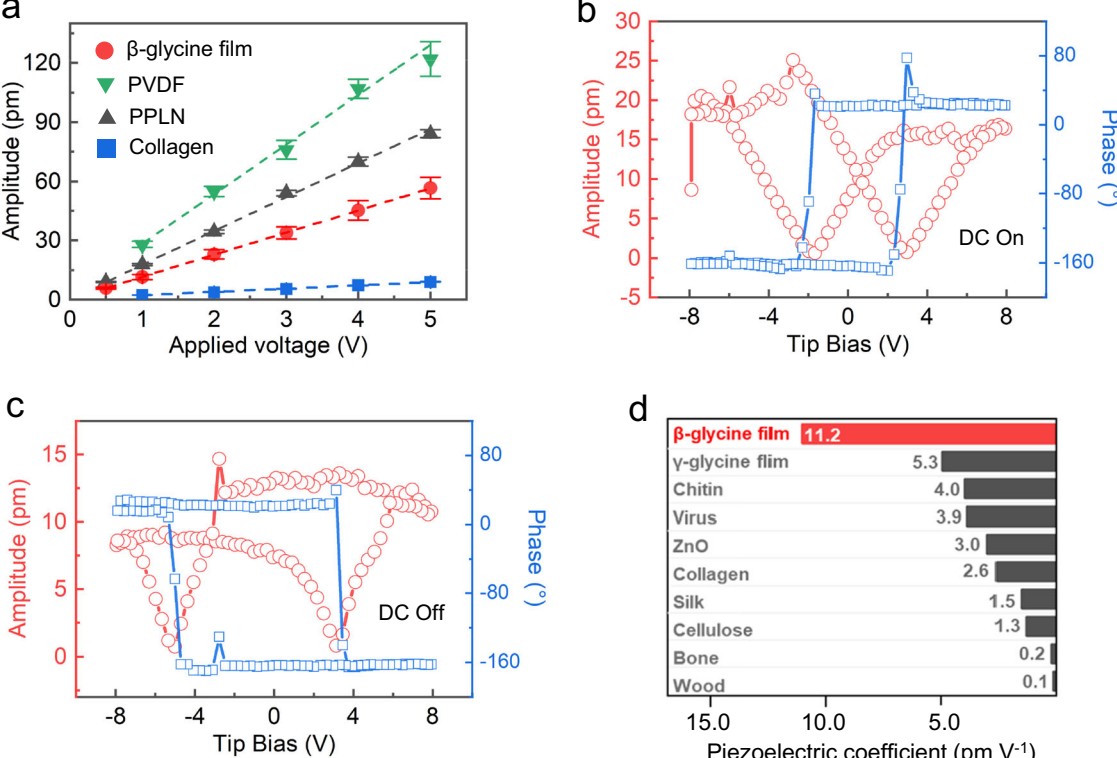

**Fig. 4 | Measurements of effective piezoelectric coefficients and ferroelectric hysteresis loops. a** Linear dependence of PFM amplitudes on the applied AC voltage. Error bar denotes the standard deviation. **b** Ferroelectric amplitude and phase hysteresis loops with the DC filed on. **c** Ferroelectric amplitude and phase hysteresis loops with the DC filed off. **d** Piezoelectric coefficient of the as-prepared β-glycine nanocrystalline films compared with other bio-organic piezoelectric materials. Measured values of the piezoelectric coefficient are listed at the end of each bar.

mapping under different applied AC voltages. The OOP amplitude increases linearly as a function of the applied AC voltage, and the slope yields the effective piezoelectric coefficient around 11.2 pm V$^{-1}$ (red curve in Fig. 4a). This value is consistent with the measured value (15 pm V$^{-1}$) of β-glycine microcrystals in previous reports[42]. It is larger than the calculated value (5.7 pm V$^{-1}$)[21] since the giant shear piezoelectricity of β-glycine (195 pm V$^{-1}$) contributes noticeably to the effective longitudinal piezoresponses when the OOP orientation of nanograins is not strictly along the longitudinal direction (2 direction) of crystallography[43,44]. To justify the measured piezoelectric coefficient using PFM, the same measurements on the periodically poled lithium niobate (PPLN), commercial polyvinylidene difluoride (PVDF) film, and collagen films are also performed for comparison (Supplementary Figs. 14–16). We then plot their piezoresponses under AC voltages that show the piezoelectric coefficients of about 17.2 pm V$^{-1}$ (PPLN), 25.2 pm V$^{-1}$ (PVDF), and 1.6 pm V$^{-1}$ (collagen) (gray curve, green curve, and blue curve in Fig. 4a), in good agreement with the value measured by quasi-static d$_{33}$ meter or reported in the previous literatures[45]. Furthermore, we perform ferroelectric hysteresis loop measurements and show that the β-glycine nanocrystalline films are also ferroelectric (Fig. 4b and Fig. 4c)[23,46]. We determine the piezoelectric coefficient of -13.3 pm V$^{-1}$ based on the saturated piezoelectric response of the amplitude hysteresis loop measured during the DC off state (Fig. 4c). This value is consistent with the linear fitting one, manifesting the effectiveness of the measured piezoelectric coefficients. In addition, we also conduct d$_{33}$ meter measurements and obtain a piezoelectric coefficient of about 11 pm V$^{-1}$, demonstrating a consistent piezoelectric property of the films at both the bulk scale and nanoscale (Supplementary Fig. 17). As shown in Fig. 4d, the piezoelectric strength of the β-glycine nanocrystalline films is superior to most reported bio-organic films (Supplementary Table 1). In this work,

the effective OOP piezoelectric value is regarded as the d$_{33}$ of the films, although the polarization direction is along the two-axis of the monoclinic β-glycine crystal. The uniform and high-value d$_{33}$ could be attributed to the excellent polarization alignment due to in-situ poling during the β-glycine homogeneous nucleation, which exposes most of the (020) polar surface toward the OOP direction.

## Thermostability and overview of piezoelectric properties

Ferroelectric materials will lose their piezoelectricity once the temperature exceeds Curie temperature (T$_C$); a high T$_C$ is preferred for applications. Unfortunately, bulk β-glycine crystals are the least stable and readily transform to α-glycine in moist air after being left at room temperature for several hours or heated to 67 °C[31]. Hence, we investigate the thermostability and phase transition properties of the as-prepared β-glycine nanocrystalline films. Measurements of differential scanning calorimetry (DSC) on the as-fabricated films reveal two thermal anomalies at roughly 192 °C and 255 °C, respectively (Fig. 5a). Combined with the thermal gravimetric analysis (TGA) results, we confirm the decomposition temperature of the glycine molecules is 255 °C. We also perform the DSC tests on γ-glycine and α-glycine crystals for comparison (Supplementary Fig. 18). To further verify whether the thermal anomaly at 192 °C is from phase transition or from melting, we conduct in-situ XRD measurements (Fig. 5b) and observe no phase other than the β phase throughout the heating process, which is also verified by the in-situ Raman measurements (Fig. 5c). The weak peak at 2θ of about 26.9 degrees in the XRD pattern corresponds to the peak of the silicon substrate used in the test, and it persists after the decomposition of the glycine molecules. The peak position shifts in the XRD pattern and Raman spectra could be attributed to the thermal expansion in the crystal lattice with increasing temperature. The temperature dependency tests of relative permittivity at different

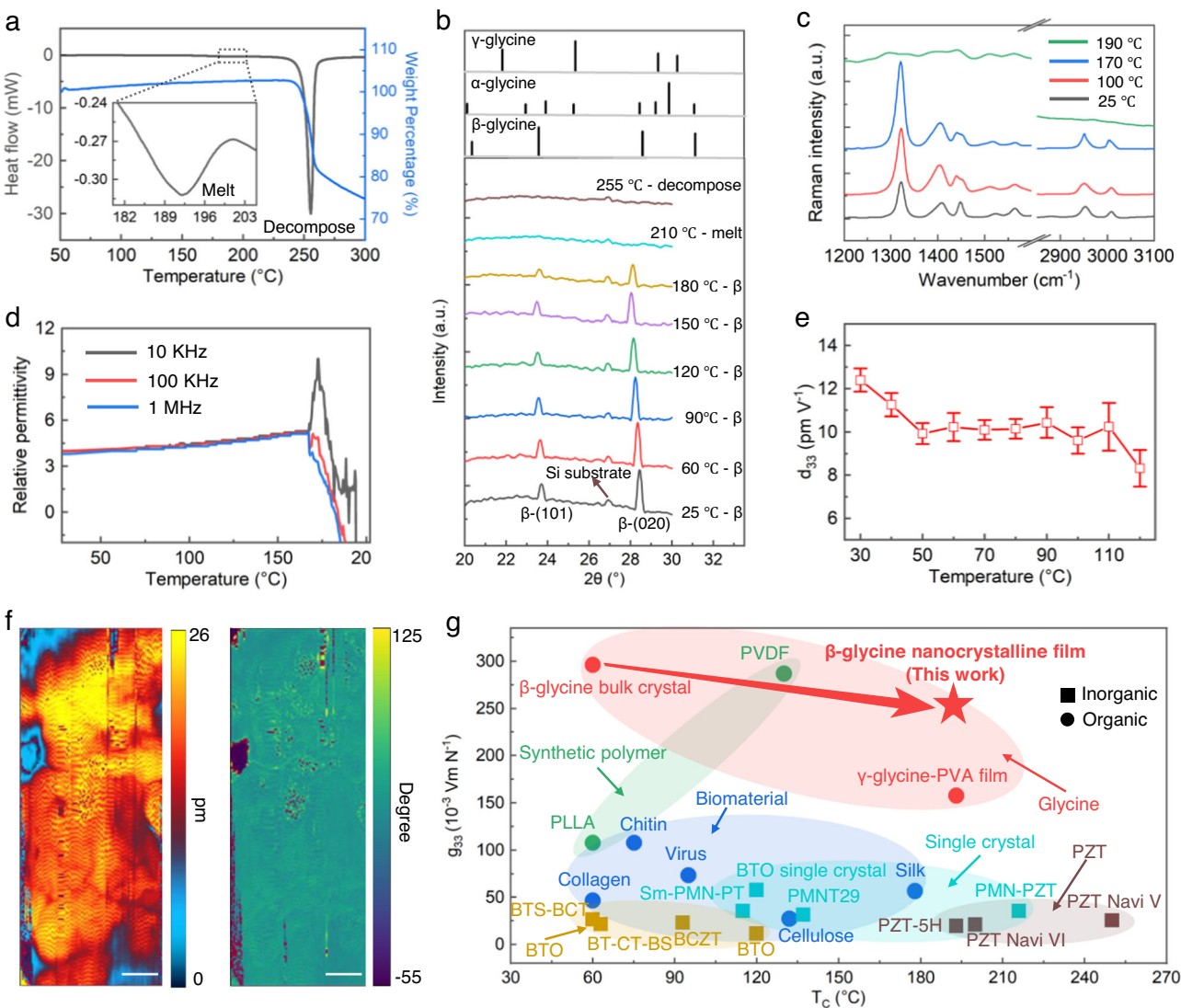

**Fig. 5 | Thermostability and overview of piezoelectric properties. a** DSC (black curve) and TGA (blue curve) results of the as-prepared films. The inset figure is the enlarged drawing of DSC curve between the temperature of 180 °C and 205 °C. **b** The in-situ variable temperature XRD patterns of the films. **c** The in-situ variable temperature Raman spectra. **d** The temperature-dependent relative permittivity for selected frequencies. **e** Dependence of d₃₃ of the β-glycine nanocrystalline films on temperature. Error bar denotes the standard deviation. **f** PFM OOP amplitude mapping (left) and phase mapping (right) under 120 °C confirming the stable piezoelectric effect of the films under high temperature. The applied AC voltage is 2 V. Scale bar: 200 nm. **g** Comparison of longitudinal piezoelectric voltage coefficient g₃₃ of most actively studied piezoelectric material systems with this work as a function of Curie temperature T_C.

frequencies also show no evident anomaly before the electrical breakdown of the films, indicating the absence of a Curie transition prior to the melting temperature (Fig. 5d). This suggests that the disappearance of the β-glycine nanocrystalline films at 192 °C is due to melting, where the melting point depression effect is expected from the nanoscale crystal size (Supplementary Note 2)[47]. We also conduct in-situ variable temperature PFM measurements and confirm the high-temperature piezoelectric effect of the β-glycine nanocrystalline films (Fig. 5e and Fig. 5f). The infinite thermostability of the films prior to the melting temperature of 192 °C is in good agreement with the observations of β-glycine crystals confined in nanopores[36,47], collectively supporting the Ostwald step rule, which posits that an otherwise metastable form of a crystalline substance, β-glycine, actually becomes the stable form when the crystal size is constrained to nanometer-scale dimensions.

In addition, we evaluate the piezoelectric voltage coefficient g₃₃ which directly represents a key material figure of merit for sensing and energy harvesting applications. The value of g₃₃ can be calculated

through the piezoelectric strain coefficients and the dielectric permittivity via the equation $g_{33} = d_{33}/\varepsilon_{33}$, in which $\varepsilon_{33} = \varepsilon_r \times \varepsilon_0$. Because of their low relative permittivity of nearly 5 (Supplementary Fig. 19), the β-glycine nanocrystalline films exhibit a superb piezoelectric voltage coefficient ($g_{33} = 252 \times 10^{-3}$ Vm N⁻¹). Based on the above results, we compare the piezoelectric voltage coefficient g₃₃ of several actively studied piezoelectric material systems together with the β-glycine nanocrystalline films as a function of T_C (Fig. 5g, Supplementary Table 2). The melting temperature of 192 °C of the β-glycine nanocrystalline films can be regarded as their equivalent T_C, which is higher than T_C of most piezoelectric materials and comparable to that of PZT-5H type piezoceramics. Notably, the g₃₃ of the β-glycine nanocrystalline films is on the same order of magnitude as PVDF, but with much higher T_C. This value is also much higher than that of most piezoelectric ceramics, inorganic piezoelectric single crystals, and piezoelectric biomaterials. The fabricated β-glycine nanocrystalline films in this research exhibit a stable structure and excellent performance over a broad temperature range, suggesting its capability as a high-

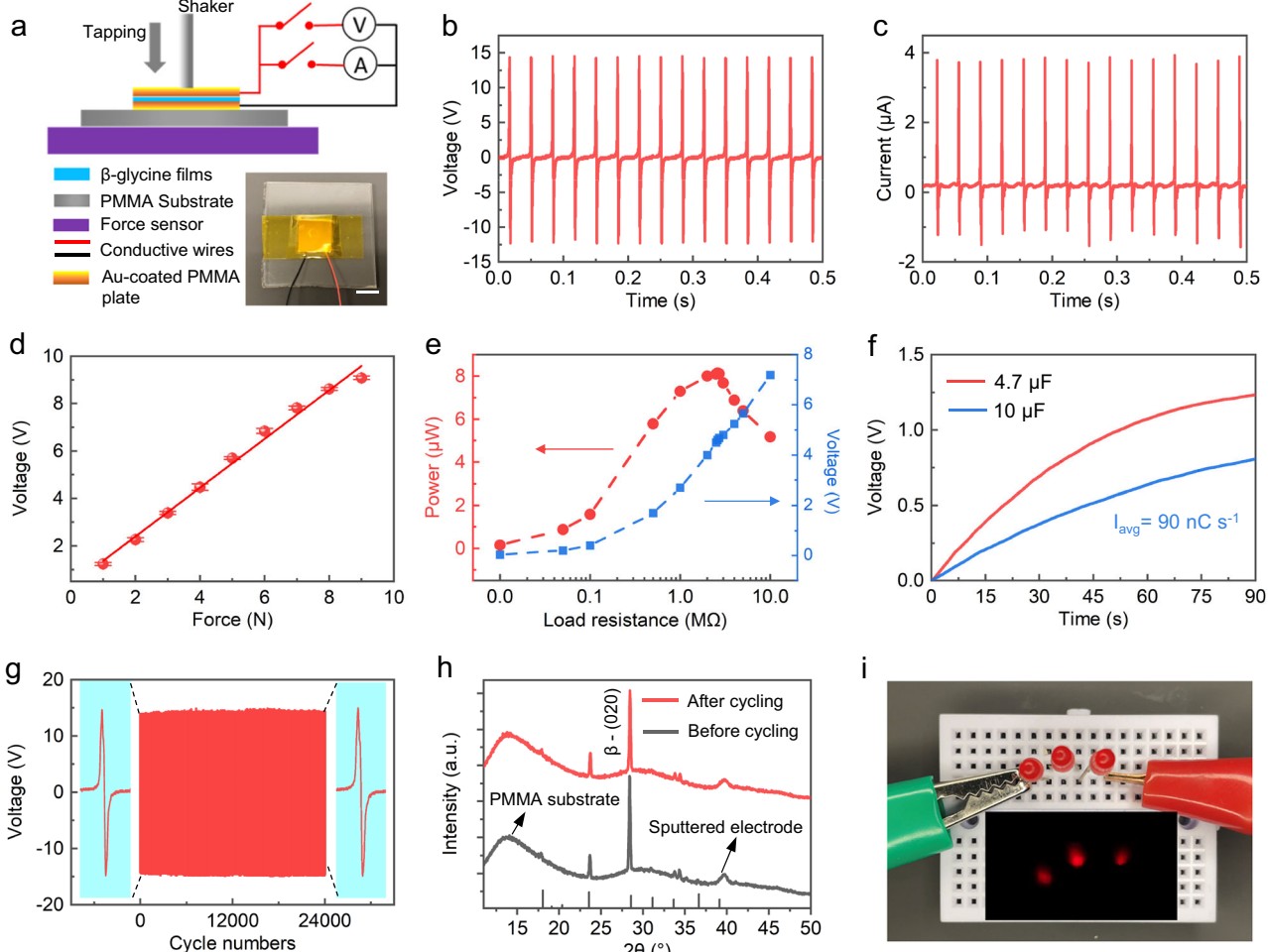

**Fig. 6 | Piezoelectric electrical performance and mechanical stability of β-glycine nanocrystalline films. a** Schematic of the piezoelectric device measurement set-up. The bottom right image is the photography of a real device. Scale bar: 1.5 cm. **b** Open-circuit voltage and **c** short-circuit current responses of the piezoelectric device of the as-prepared films under 1.5 MPa compressive pressure with 30 Hz frequency. **d** Linear dependence of the maximum open-circuit voltage output of the piezoelectric devices on the increased external forces. Error bar denotes the standard deviation. **e** Dependence of the power output of the piezoelectric device on the resistance of the external load under 1.0 MPa compressive pressure. **f** The charging curves of the 4.7 µF and 10 µF capacitor. **g** Voltage output signals during 24,000 cycles under 1.5 MPa compressive pressure. **h** XRD spectra of the films before and after durability tests of 24,000 compressing cycles. The standard XRD spectra of β-glycine are shown at the bottom of the figure. **i** Optical image showing three LEDs lit up by an individual piezoelectric device through finger tapping with a frequency of 0.5–1 Hz.

performance piezoelectric sensor attractive for various applications. Our design strategy of constructing nanocrystals and aligning the domains by in-situ poling in the biomolecular thin films has the advantage of improving the piezoelectric coefficients while simultaneously maintaining a high equivalent $T_C$.

## Piezoelectric outperformance and mechanical stability characterizations

To intuitively evaluate the piezoelectric performance, the electric output of the films is characterized under impulse forces. The β-glycine nanocrystalline films are deposited on a gold-coated polymethyl methacrylate (PMMA) substrate ($1.5 \times 1.5$ cm²) and sandwiched with another conductive PMMA plate. The characterization set-up including a shaker with a force sensor is shown in Fig. 6a. A compressive force is repeatedly applied to the sandwiched device over an area of 10 mm² at different frequencies. The β-glycine nanocrystalline film produces a maximum open-circuit voltage of about 14.5 V (Fig. 6b, and control devices are shown in Supplementary Figs. 20, 21), and a maximum short-circuit current of nearly 4 µA (Fig. 6c), which are an order of magnitude larger than most reported piezoelectric biomaterials (Table s3). The signs of the output voltage all agree with the

polarization direction, which aligns along the applied electric field in the synthesis process (Supplementary Fig. 22). The reversed connection tests show reversed outputs (Supplementary Figs. 22, 23), and the open-circuit voltage also exhibits excellent linearity with forces ranging from 1 N to 9 N (Fig. 6d), again confirming that the measured electrical signal is indeed from the piezoelectricity.

The β-glycine nanocrystalline films generate a power density of up to 3.61 µW cm⁻² connecting to a load resistor of 2.6 MΩ, which is one to three orders of magnitude higher than other piezoelectric biomaterials-based power generators (Fig. 6e, Supplementary Table 3). The high power density makes it a promising alternative to other renewable energy harvesting methods such as solar, geothermal, wind, hydro, thermoelectric, and biomass (Supplementary Table 4). Utilizing a full-bridge rectifier to rectify the generated AC signals to DC signals (Supplementary Fig. 24), the voltage stored in capacitors (4.7 µF and 10 µF) increases to 1.24 V, and 0.81 V, respectively, in a nighty-second charging process (Fig. 6f). The average charging rate for a 10 µF capacitor is calculated to be 90 nC s⁻¹, which indicates the capability of the β-glycine nanocrystalline films as a stable power supply for implantable devices. After 24,000 compressing cycles, the open-circuit voltage output remains almost unchanged (Fig. 6g). The XRD

and SEM results confirm neither phase transition nor mechanical damage after cycling (Fig. 6h and Supplementary Fig. 25), manifesting the high durability and reliability of the films. The high output performance and property uniformity allow an individual device to light up three LEDs simultaneously (Fig. 6i and Supplementary Movie 2), which to our best knowledge, is the first time a piezoelectric bio-organic film has been achieved. The high output performance could be attributed to the optimal polar orientation in the OOP direction and the compact, dense polycrystalline structure of the single-component biomolecular films.

## Discussion

This work addresses the long-standing challenge of synthesizing large-scale high-performance piezoelectric biomaterials. We have developed a generalizable route to fabricate customizable bio-organic piezoelectric thin films via the synergy of nanoconfinement and in-situ poling. The nanoconfinement, along with the homogeneous nucleation, enables the large-scale out-of-plane (OOP) alignment of crystal grains in the strongest polarization direction by in-situ electric field, resulting in superb uniform piezoelectricity and high thermal stability. It should be noted that only a fraction of giant shear piezoelectricity (195 pm V$^{-1}$) has been exploited, and the β-glycine nanocrystalline films hold the potential of achieving higher piezoelectric performance by rational structural design. The excellent output performance, natural biocompatibility, and biodegradability of the β-glycine nanocrystalline films are of practical implications for high-performance transient biological electromechanical applications. As a paradigm shift toward commonly used bottom-up self-assembly methods, our approach is extricated from the interface dependency due to the homogeneous nucleation of β-glycine nanocrystals. It is scalable to create films with variable dimensions, programmable structures, and diverse material forms such as flexible composites. Furthermore, this strategy can be applied to design large-scale films of various biomaterials and other piezoelectric materials, such as molecular or organic-inorganic piezoelectric materials.

## Methods

### Construction of the bio-organic film printer

The printer comprises a stainless-steel nozzle with an outer/inner diameter of 0.31/0.16 mm, a syringe pump ((LSP01-2A, Baoding Longer Precision Pump Co., Ltd., China) used to supply the glycine solution to the nozzle, an X-Y translational stage for mounting substrates, and a power supply (DW-P303-1ACH2, Dongwen high voltage power supply Co., Ltd, China) connected between the nozzle and the stage for applying an electric field. The computer-controlled translational stage is electrically ground well to dissipate the electrostatic charge of printing substrates fast enough. Upon application of a high voltage, a charged jet is expelled from the nozzle and broken into tiny droplets at the end of jet flow. The size of the atomized droplets depends on the distance between the nozzle and the substrate, the applied voltage, the supply rate, and the solution properties. For the glycine film deposition, the distance between nozzle and substrate is 2–6 mm, and the applied voltage is typically 3–5 kV. The flow rate of the solution is 0.2–0.5 μL min$^{-1}$.

### Fabrication of β-glycine nanocrystalline films

The glycine solution is prepared by dissolving 2 g glycine powder (Macklin/AR9, 9.5–100.5%) in 20 ml deionized water (10% w/v), and then the mixture is stirred for 3 h at 60 °C using magnetic rotator to obtain a homogeneous solution. The as-obtained glycine solution is then directly used for film synthesis. At room temperature, the solution is applied to a small syringe of the bio-organic films printer. During the electrohydrodynamic spray process, the electric field overcomes the surface tension of the glycine aqueous solution to produce numerous nanodroplets, leading to the formation of nanocrystals. The

depositing speed $1.5 \times 10^7$ μm$^3$ s$^{-1}$ is determined under the flow rate (0.5 μL min$^{-1}$) and designated X-Y translational stage movement route (Supplementary Fig. 26). Through regulating the operation time and controlling the X-Y translational stage movements, the piezoelectric bio-organic thin films with variable thicknesses and diverse sizes are finally formed on the gold-coated substrate.

### Structural characterizations

The X-ray diffraction (XRD) pattern is acquired by a wide-angle X-ray diffractometer (PANalytical X'pert3 diffractometer) using Cu Kα radiation of wavelength 1.54060 Å. The XRD data are collected at room temperature in the range of 10–50° (2θ) using a step size of 0.02°. Structural compositions of the films are obtained using Raman spectroscopy (Thermo Fisher DXR2xi, laser source of λ = 532 nm and power of 5 mW). The microstructures of the surfaces and cross-sections of the films are observed with scanning electron microscopy (SEM; FEI Quanta 450). Before the SEM measurements, all samples are coated with the gold electrode by magnetron sputtering (Q150TS).

### Vertical PFM measurements

To determine the OOP piezoelectric responses of the samples, DART mode is used to reduce noise and topography crosstalk based on an Asylum Cypher ES AFM system. For all PFM measurements and SKPM measurements in this work, a conductive probe (Nano world Arrow-EFM) with Cr/PtIr coating on both cantilever and tip is used. The nominal resonance frequency and the nominal stiffness of the probe are 75 kHz and 2.8 N m$^{-1}$, respectively. Before all PFM measurements, both the inverse optical lever sensitivity and spring constant are calibrated using Asylum's software GetReal. The contact resonance frequency is typically around 300 kHz. To calculate the effective piezoelectric coefficient $d_{33}$ of the β-glycine nanocrystalline films, a small area of $300 \times 300$ nm$^2$ is scanned under AC voltages between 0.5 and 5 V. The measured ten areas are randomly selected from two films on the gold-coated silicon substrate ($1 \times 1$ cm$^2$). The intrinsic piezoresponses are obtained by correcting the resonance amplification with the quality factor using the Calc SHO Parms function in the DART panel. The average vibration amplitudes in each scanned area are recorded for linear fitting to calculate the effective $d_{33}$. The error bar denotes the standard deviation between the average amplitudes of ten selected areas. For the $d_{33}$ calculation of the PPLN sample, an area ($20 \times 20$ μm$^2$) is scanned under AC voltages between 0.5 and 5 V. The effective $d_{33}$ is determined by linear fitting the average amplitudes in a selected small area ($4 \times 4$ μm$^2$). The error bar denotes the standard deviation of each mapping under different AC voltages. Ferroelectric hysteresis loops are measured by applying DC voltage sweeps from –8 to +8 V at 1 V AC drive voltage using DART-SS-PFM.

### Lateral PFM measurements

To determine the IP piezoreponses, the lateral PFM measurements are conducted with a BRUKER Dimension Icon system in the vertical and horizontal PFM mode. For the characterization of effective shear piezoelectric coefficients of collagen films, an area ($2 \times 2$ μm$^2$) is scanned under the applied voltage amplitude on the tip varying from 2 V to 10 V. The IP piezoelectric signals are recorded and averaged over the scanned small area to calculate the piezoelectric coefficients. The error bar stands for the standard deviation of each mapping under different AC voltages. In order to be consistent with the scale of the β-glycine nanocrystalline films, the applied voltage and the corresponding amplitude of the collagen films are reduced by two times.

### In-situ PFM measurements

The in-situ variable temperature PFM measurements are performed in DART mode using an Asylum Cypher ES AFM system. The

temperature is controlled in the environment panel with a CoolerHeater Stage. The PFM mappings are obtained at different temperatures (30, 40, 50, 60, 70, 80, 90, 100, 110, and 120 °C). The heating rate and scanning rate are set as $1\,°C\,s^{-1}$ and 0.7 Hz, respectively. When heated to the target temperature, it will be maintained for 3 min. The average amplitudes in five small areas of $300 \times 300\,nm^2$ under the applied AC voltages of 2 V are recorded to calculate the effective piezoelectric coefficient under different temperatures. The error bar represents the standard deviation between the average amplitudes of five selected areas.

## Bulk-scale $d_{33}$ and relative permittivity measurements

The piezoelectric charge constants are measured with a quasi-static d33 meter (YE2730A, Sinocera Piezotronics Inc.). Temperature dependence of the relative permittivity is examined by a precision LCR meter (E4980A, Agilent Technologies) connected to a computer-controlled temperature chamber.

## Thermostability measurements

Differential scanning calorimetry (DSC) results are collected using a DSC 3 (METTLER TOLEDO) differential scanning calorimeter at a scan rate of $10\,°C\,min^{-1}$. An indium standard is used to calibrate the instrument, and argon is used as the purge gas. Thermal gravimetric analysis (TGA) is performed on a TGA/DSC 3+ (METTLER TOLEDO), and the scan rate is $10\,°C\,min^{-1}$. The scan ranges of DSC and TGA are 25−300 °C. The in-situ variable temperature XRD measurements are performed using a Rigaku Smartlab using Cu Kα radiation of wavelength 1.54060 Å, collecting XRD data for 3 min at each temperature (25, 60, 90, 120, 150, 80, 210, 230, and 255 °C). The heating rate and scanning speeds are $10\,°C\,min^{-1}$ and $5°\,min^{-1}$, respectively.

## Piezoelectric device fabrication

Two PMMA plates ($1.5 \times 1.5\,cm^2$) are deposited with gold electrodes using the magnetron sputtering machine (Q150TS). The β-glycine nanocrystalline films are directly formed on the gold-coated PMMA plate using the bio-organic films printer. Then the PMMA plate with the β-glycine film is fixed to a $4 \times 4\,cm^2$ thicker PMMA plate for convenient handling. Another bare gold-coated PMMA plate is put on top of the β-glycine film by PI tape to serve as the top electrode. Two conductive wires are connected to two gold electrodes using the silver paste to complete the device fabrication. By replacing the film with a non-piezoelectric Kapton PI tape or commercial PVDF films, we designed the control devices with the structural assembly and measurement parameters consistent with the β-glycine film-based piezoelectric devices.

## Piezoelectric device characterization

The piezoelectric device of the β-glycine films is mounted to a fixture, and a vertical compressing force is applied by a vibration generator, with a controllable oscillation frequency and adjustable force by changing the distance between the compressing pillar and sample surface. The applied force is detected and quantified by a mechanical force sensor. The voltage outputs and current outputs of the piezoelectric device are recorded using a digital oscilloscope (Rohde & Schwarz RTE1024) and a low-noise current preamplifier (Stanford Research SR570), respectively.

## Data availability

The data that support the findings of this study have been included in the main text and Supplementary Information. All other relevant data supporting the findings of this study are available from the corresponding authors upon request.

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

## Acknowledgements

The work described in this paper is supported by General Research Grant (Project Nos. 11212021, 11210822 (Z. Y.)) and Early Career Scheme (Project No. 21210619 (Z. Y.)) from the Research Grants Council of the Hong Kong Special Administrative Region. We thank CityU technician Mr. Kwok Cheong for the technical support and Ms. Yiwe Hu from Shiyanjia Lab for in-situ XRD and in-situ Raman measurements.

## Author contributions

Z.Z. and X.L. contributed equally to this work. Z.Z., X.L., and Z.Y. conceived the research. Z.Z. and X.L. designed the experiments and fabricated the samples. Z.Z. performed the structural characterizations, piezoelectricity measurements, thermostability measurements, and device testing and analyzed the data with assistance from X.L., Z.P., X.Y., S.L., Y.H., X.X., Y.S., L.J., and B.L. Z.P. conducted the XRD measurements. X.Y. conducted the dielectric measurements. S.L. and Y.H. helped perform the ferroelectric hysteresis loop measurements. Y.S. helped on the bulk-scale d33 measurements. X.X. assisted in PVDF characterizations. L.J. assisted in device testing. B.L. conducted the TGA measurements. X.Z. assisted in operating the Cypher AFM, and is supervised by Y.C. Z.Z. drafted the paper. Z.Y. supervised the study. Z.Z., X.L., S.Z., A.J., and Z.Y. reviewed and revised the paper with input from all authors.

## Competing interests

Z. Y., Z. Z., and X. L. declare there is a potential competing interests: United States Non-Provisional Patent Application No. 17/661,925 Title: PIEZOELECTRIC BIO-ORGANIC FILMS AND FABRICATION METHOD THEREOF Assignee: City University of Hong Kong Inventors: (1) Z.Y.; (2) Z.Z.; and (3) X.L. Filed on 4 May 2022. The remaining authors declare no competing interests.
