## [Peer Review File · Nature Communications]

Active self-assembly of piezoelectric biomolecular films via synergistic nanoconfinement and in-situ polingREVIEWER COMMENTS

Reviewer #1 (Remarks to the Author):

The authors showed thermal stability and phase-stabilisation of meta-table beta glycine, which only a few groups in the world have done so far. Furthermore, they showed that in situ poling of these films and demonstrated a power density of over 3.5 micro watt, which brings it on a par with what can be obtained with solar power. I think, this work is worthy to be published in Nature Communication but the authors need to highlight power density comparisons with respect to other energy harvesting options from renewable sources such as solar, thermoelectric, wind, geothermal etc. The authors should also highlight the fact that they realised only a fraction (10%) of the reported strain constant (~ 180 pC/N) of beta glycine. Overall, I recommend publishing this work as it truly represents a clear progress beyond state of the art, will be of general interest to many and written in reasonably good style to attract wider readership.

Reviewer #2 (Remarks to the Author):

The paper by Zhang et al. reports a novel technique for the fabrication of polycrystalline films of β -glycine, the least stable but the most interesting phase for the applications. Though the formation of the oriented film is fairly demonstrated, its characterization raises many doubts.

1. The authors report the thickness of the film may vary from 0.6 to 9 μm . However, they do not indicate the parameters controlling its thickness.
2. The authors write (lines 120–122): “The in-situ electric field in the crystal growth process induces the domain alignment of β -glycine nanocrystals, suggesting the net polarization direction [020] is parallel to the electric field (Fig. 1d).” However, Figure 1d is an illustration only and cannot serve as a proof of this statement. The co-direction of the polarization and the electric field should be experimentally demonstrated. That could be done, for instance, by polarized Raman spectroscopy. Angular dependencies of intensities of specific spectral lines can provide direct information on the molecular orientation in the film. A similar analysis has been done in ACS Appl. Mater. Interfaces 9, 20029–20037 (2019) (doi: 10.1021/acsami.7b02952).
3. Though the effect of the nanoconfinement and electric field on the predominant crystallization is confirmed by the powder XRD diffractograms, the mechanism suggested seems to be too schematic. The authors write (lines 122–123): “The crystallization is mostly completed before depositing on the substrate, while numerous β -glycine nanocrystals still carrying a thin water film...” whereas it is known

that the crystallization starts from the edges of the droplet (independently whether droplet at the substrate or not), where the supersaturation state is achieved due to higher solvent evaporation and Marangoni flows. Then, the front of crystallization propagates to the droplet bulk. Therefore, the deposition scheme suggested by the authors, where the crystalline core is covered by the liquid shell, is unlikely.

4. The estimation of piezoelectric coefficients by the authors is also doubtful. The authors declared the d_{33} value of 11.2 pm V⁻¹ and wrote that “this value is consistent with the measured value (15 pm V⁻¹) of β -glycine microcrystals in previous reports³⁵ but larger than the calculated value (4.7 pm V⁻¹)²¹ since the giant shear piezoelectricity of β -glycine (178 pm V⁻¹) also contributes to the OOP piezoresponses^{36,37}”. This one sentence contains some errors.

a. First, the presented values of d_{33} and a “giant shear piezoelectricity” for β -glycine calculated in Ref. 21 are incorrect. The correct values from that paper are 5.7 and 195 pm V⁻¹, respectively.

b. Second, the “giant shear piezoelectricity”, obviously related to the d_{16} piezoelectric coefficient, cannot contribute to the out-of-plane (OOP) signal detected by the authors. Following equation (1) from Ref. 37, the OOP response includes only coefficients d_{31} , d_{15} , and d_{33} . The coefficient d_{16} cannot contribute to the vertical deformation of the crystal under an electric field.

c. Since the electromechanical response in PFM measurements is highly influenced by electrostatic interactions of non-piezoelectric origin, crystals with well-known piezoelectric tensors are typically used to calibrate the measured values. Lithium niobate is one of the most studied piezoelectric crystals with the table value of $d_{33} = 6.2$ pm V⁻¹ [Opt. Mater. 23, 403-408 (2003), doi: 10.1016/S0925-3467(02)00328-2], whereas the authors measured the raw value 17.2 pm V⁻¹. Therefore, a correction factor of $6.2/17.2 = 0.36$ should be used to get the correct values of d_{33} . Taking into account this factor, for β -glycine, the $d_{33} = 11.2 \times 0.36 = 4.03$ pm V⁻¹, which is a more reliable value comparable with that predicted in Ref. 21.

d. The recalculation of d_{33} considering the correction factor will also lead to the reduction of the piezoelectric voltage coefficient g_{33} , which is will be 90×10^{-3} VmN⁻¹ (and it is not the record value anymore).

5. However, the main doubts are related to energy harvesting measurements. Though β -glycine indeed possesses high piezoelectric properties, it is a very fragile material demonstrating a tendency to transition into more stable phases not only under heating but mechanical stress as well [J. Therm. Anal. Calorimetry 73, 419–428 (2003), doi: 10.1023/A:1025457524874]. While measuring response under stress, the authors applied very high forces comparable with the action of a hummer drill (forces up to 9 N with frequency up to 40 Hz), and the film survived 24000 cycles without any degradation! Moreover, the film demonstrated unconceivable values of short-circuit current and output power of 3-6 orders of magnitude higher than in any other material (Table S3)! That looks fantastic and very unlikely! I am afraid it could be an error in the calculations and/or in the measurements and should be double/ or triple-checked. Did the authors see any mechanical damage after cycling?

6. A little correction for the thermal measurements. The authors wrote (line 204–205): “we confirm the decomposition temperature of β -glycine nanocrystalline films is 255 °C”. However, this is incorrect. The

nanocrystalline film stops existing at the moment of melting at 192 °C. At 255 °C, the glycine molecules evaporation probably takes place.

7. Is there any sign of ferroelectricity in these films? What is the phase transition temperature? Is there any sign of Curie-Weiss behavior in the dielectric constant? These complementary data may give some clue for the unusually high energy harvesting figures of merit.

8. The section on DL-threonine films and crystals is totally abundant. It takes only one paragraph and does not report any essential results. This part should be completely deleted.

To summarize, the paper by Z. Zhang et al. is not ready for publication in its current form and needs additional proof and reconsideration of the experimental results. Nevertheless, the suggested technique is promising and may be of great interest to readers of a more specialized journal.

Reviewer #3 (Remarks to the Author):

This work reports a synthesis of β -glycine film by the electrohydrodynamic spray method. Electric field applied during the spray was claimed being able to lead the formation of β -glycine and align the piezoelectric polarization. The as-synthesized film showed a strong piezoelectricity, and retained it up to 192 °C. The results are interesting and might be impactful as a new piezoelectric bio-crystal thin film. However, there are some key factors that are defective, which undermine the significance of this work.

The most important issue is the phase and associated good thermostability. It is known that the β -glycine is not stable at high temperature, which is also highlighted by the authors as the main novelty. However, this important advancement is not explained. Why this polycrystalline film can stabilize the beta phase? Either thermodynamically or kinetically, it does not make sense simply by forming such a film, unless the phase itself is stable. Authors need to carefully analyze the XRD spectra, as there were only a few peaks shown. The strong peak may also come from other phases such as the gamma phase. As temperature increases, the strong peak also shifted to lower angle. Why? In order to support that it is the true beta phase property, authors need to carefully confirm the phase with more detailed diffraction analysis on peak positions for different phases, and provide a reasonable explanation why this film or approach can substantially improve the beta-phase stability. The explanation should also be supported by control experiments to validate the hypothesis.

The d_{33} was only measured by a PFM. This is not reliable and not adequate to claim the piezoelectric property of the large and thick films. PFM can easily be influenced by many surface features, such as large roughness or edge effect, and can show substantially different location-dependent results.

Measuring only a very small area and use it to represent the entire film's property is not acceptable. Authors should provide bulk scale measurements to confirm the film's piezoelectric property.

Piezoelectric and ferroelectric were both used in the paper randomly. It is not accurate. There is no ferroelectric property shown at all in the work. When talk about polarization alignment, piezoelectric refers to crystal orientation, and ferroelectric refers to ionic dipole orientation.

Aside from the defective materials synthesis and characterization approach and discussion, the application demonstration is routine and simple. No unique application that relies on the biomaterials nature is demonstrated, which make this work less attractive.

Reviewer Comments (and change made in accordance)

Reviewer #1 (Remarks to the Author):

The authors showed thermal stability and phase-stabilisation of meta-table beta glycine, which only a few groups in the world have done so far. Furthermore, they showed that in situ poling of these films and demonstrated a power density of over 3.5 micro watt, which brings it on a par with what can be obtained with solar power. I think, this work is worthy to be published in Nature Communication but the authors need to highlight power density comparisons with respect to other energy harvesting options from renewable sources such as solar, thermoelectric, wind, geothermal etc. The authors should also highlight the fact that they realised only a fraction (10%) of the reported strain constant (~ 180 pC/N) of beta glycine. Overall, I recommend publishing this work as it truly represents a clear progress beyond state of the art, will be of general interest to many and written in reasonably good style to attract wider readership.

Response: We are pleased that the referee acknowledged the novelty and significance of our research. Thank you. Following the referee's insightful suggestions, we have compared the power density with other energy harvesting technique and highlighted the potential of exploiting the giant shear strain constant (~ 195 pC/N) in the revised manuscript (the revised portions are marked in Red Color in the manuscript) as below.

Results

The β -glycine nanocrystalline films generate a power density of up to $3.61 \mu\text{W cm}^{-2}$ connecting to a load resistor of $2.6 \text{ M}\Omega$, which is one to three orders of magnitude higher than other piezoelectric biomaterials-based power generators (Fig. 6e, Table s3). The high power density makes it a promising alternative to other renewable energy harvesting methods such as solar, geothermal, wind, hydro, thermoelectric, and biomass (Table s4).

Discussion

The nanoconfinement, along with the homogeneous nucleation, enables the large-scale out-of-plane (OOP) alignment of crystal grains in the strongest polarization direction by in-situ

electric field, resulting in superb uniform piezoelectricity and high thermal stability. It should be noted that only a fraction of giant shear piezoelectricity (195 pm V^{-1}) has been exploited, and the β -glycine nanocrystalline films hold the potential of achieving higher piezoelectric performance by rational structural design.

Table S4: Comparison of the power densities of this work with other renewable energy harvesting methods.

Type	Power density ($\mu\text{W cm}^{-2}$)	Reference
Solar	663	Energy Policy 123 , 83–91 (2018)
Geothermal	224	Energy Policy 123 , 83–91 (2018)
Wind	184	Energy Policy 123 , 83–91 (2018)
Hydro	14	Energy Policy 123 , 83–91 (2018)
Thermoelectric	40	Metrology and Measurement Systems 23 , 495–512 (2016).
Biomass	8	Energy Policy 123 , 83–91 (2018)
Electromagnetic	4	Metrology and Measurement Systems 23 , 495–512 (2016).
This work (mechanical energy harvesting)	3.61	

Reviewer #2 (Remarks to the Author):

The paper by Zhang et al. reports a novel technique for the fabrication of polycrystalline films of β -glycine, the least stable but the most interesting phase for the applications. Though the formation of the oriented film is fairly demonstrated, its characterization raises many doubts.

1. The authors report the thickness of the film may vary from 0.6 to 9 μm . However, they do not indicate the parameters controlling its thickness.

Response: Thank the referee for the comments. We determined the depositing speed of the films based on designated fabrication parameters. We have added the description in the Results section and given the detailed calculation method in the Method section of the revised manuscript as below.

Results

The cross-section scanning electron microscopy (SEM) image and surface topography SEM image show the uniform and compact nanograin distribution of β -glycine films with thicknesses ranging from 0.6 to 9 μm with the depositing speed of $1.5 \times 10^7 \mu\text{m}^3 \text{s}^{-1}$ (Fig. 2a, Fig. 2b, Fig. s4, Fig. s5, and methods section), where the average grain size is approximately 200 nm (Fig. 2c).

Methods

Fabrication of β -glycine nanocrystalline films: The depositing speed of $1.5 \times 10^7 \mu\text{m}^3 \text{s}^{-1}$ is determined under the flow rate ($0.5 \mu\text{L min}^{-1}$) and designated X-Y translational stage movement route (Fig. s27). Through regulating the operation time and controlling the X-Y translational stage movements, the piezoelectric bio-organic thin films with variable thicknesses and diverse sizes are finally formed on the gold-coated substrate.

Figure S27. The X-Y translational stage movement route with a moving speed of 82 cm min^{-1} for determining the β -glycine film depositing speed.

2. The authors write (lines 120–122): “The in-situ electric field in the crystal growth process induces the domain alignment of β -glycine nanocrystals, suggesting the net polarization direction [020] is parallel to the electric field (Fig. 1d).” However, Figure 1d is an illustration only and cannot serve as a proof of this statement. The co-direction of the polarization and the electric field should be experimentally demonstrated. That could be done, for instance, by polarized Raman spectroscopy. Angular dependencies of intensities of specific spectral lines can provide direct information on the molecular orientation in the film. A similar analysis has been done in ACS Appl. Mater. Interfaces 9, 20029–20037 (2019) (doi: 10.1021/acsami.7b02952).

Response: We thank the reviewer for the constructive comments. Our previously conducted XRD measurements showed that the OOP orientation is dominated by being along the net polarization direction [020] according to the strongest peak (020) in XRD spectra. Furthermore, the PFM phase results also demonstrated that the ferroelectric domains of the as-prepared films are well aligned, and the polarization direction of the nanograins is consistent.

In Ref (ACS Appl. Mater. Interfaces 2017, 9, 23, 20029–20037), Ensieh Seyedhosseini et al. for the first time conducted an advanced study on the β -glycine microcrystals to probe the molecular orientation using the polarized Raman spectroscopy. As suggested by the reviewer, we performed the same measurements on our β -glycine films. We found that the spectral line $\nu(\text{C-C})$ at about 891 cm^{-1} is observed in both directions, indicating the c-axis is parallel to the in-plane of the films. The line at around 1039 cm^{-1} corresponds to $\nu(\text{C-N})$ vibrations and is observed in both spectra, thus demonstrating that the a-axis is also parallel to the in-plane of

the films. The peak intensities in the two spectra are not significantly different because the as-fabricated polycrystalline film is isotropic in the plane. The a-axis and c-axis are randomly distributed in the plane, while the b-axis is aligned in the out-of-plane direction due to the in-situ electric field.

Figure R1. Polarized Raman measurements of the β -glycine nanocrystalline films.

3. Though the effect of the nanoconfinement and electric field on the predominant crystallization is confirmed by the powder XRD diffractograms, the mechanism suggested seems to be too schematic. The authors write (lines 122–123): “The crystallization is mostly completed before depositing on the substrate, while numerous β -glycine nanocrystals still carrying a thin water film...” whereas it is known that the crystallization starts from the edges of the droplet (independently whether droplet at the substrate or not), where the supersaturation state is achieved due to higher solvent evaporation and Marangoni flows. Then, the front of crystallization propagates to the droplet bulk. Therefore, the deposition scheme suggested by the authors, where the crystalline core is covered by the liquid shell, is unlikely.

Response: We thank the referee for the professional comment. This question is fundamental and critical. Many researchers have explored the heterogeneous nucleation mechanism at the solid-liquid interface. When a droplet has a low contact angle on a substrate, the capillary forces fix the boundaries of the liquid droplet, resulting in a flow of liquid from the interior to the edge due to evaporation near the edge. This flow effectively transports all solutes to the edge, a phenomenon commonly known as the coffee-ring effect. Conversely, in droplets with a large contact angle and clean surfaces, where evaporation induces thermal Marangoni flows, excessive flow can actually redistribute solute particles back to the center of the droplet. This causes nucleation to occur predominantly at the center of the droplet rather than the edge (*Cryst. Growth Des.* 2019, 19, 7, 3869–3875; *J. Phys. Chem. B* 2006, 110, 14, 7090–7094).

However, in our work, the β -glycine nanocrystals are expected to be formed by homogeneous

nucleation due to the small size and substrate-free property of in-flight nano-micro droplets (*The Journal of Chemical Physics*, 1999, 111(14): 6521-6527, *The Journal of Physical Chemistry A*, 2005, 109(11): 2540-2546, *The Journal of Physical Chemistry B*, 2013, 117(35): 10241-10249.). According to classical homogeneous nucleation models, nucleation sites should be far from the surface, and the probability of nucleation at each point should be the same. As suggested by the study of nucleation of levitated droplets with net charge in the Ref (*Anal. Chem.* 2005, 77, 10, 3189–3197), nucleation may initiate in the diffuse layer at the droplet–air interface; however, it is also possible that it took place in the droplet core and that nuclei migrated to the diffuse layer at the droplet–air interface prior to undergoing growth. Although the nucleation site remained unclear, it was observed that further growth of the precipitate allowed it to branch away from the droplet–air interface into the interior of the levitated droplet (Figure R2 reproduced from Ref (*Anal. Chem.* 2005, 77, 10, 3189–3197) is shown below). In addition, it has been well studied that the partially wet particles with incomplete evaporation deposited on the substrate are essential for further dense film formation in the electrospray deposition techniques (*Journal of Materials Chemistry*, 1996, 6(5): 765-771, *ACS Appl. Polym. Mater.* 2023, 5, 3, 1797–1809, *Anal. Chem.* 2005, 77, 10, 3189–3197). Therefore, the crystalline core covered by the liquid shell described in our work is reasonable and necessary.

Figure R2. (A, B) Representative images of levitated droplet residues that had (A) -135 and (B) -325 fC of net charge as observed with optical microscopy. Droplets were dispensed from a starting solution containing 285 mM NaCl in water/glycerol (97:3 v/v). (C) Residues of droplets dispensed from a starting solution having 69 mM NaCl in water/glycerol (97:3 v/v) that had either -135 or -350 fC of net charge as a function of their levitation time. Reproduced from Ref (*Anal. Chem.* 2005, 77, 10, 3189–3197).

In response to the reviewer's comments, we have described more detailly the possible nucleation mechanism for nano-micro droplet crystallization in the revised manuscript. Nevertheless, further analysis and verification of the mechanism are required. Observing the nucleation and crystallization process, especially at the nanoscale, has been recognized as a significant challenge in the research community for decades. We appreciate the valuable comments from the reviewer and will explore suitable methods to investigate the entire nucleation and film-forming process in future work.

Results

We then propose the potential nucleation and crystallization process as follows. The β -glycine nanocrystals are formed through homogeneous nucleation owing to the small size and substrate-free property of in-flight nano-micro droplets (*The Journal of Chemical Physics*, 1999, 111(14): 6521-6527, *Materials Today* 24, 17–25 (2019)). As homogeneous nucleation is not influenced by solid-liquid interfaces, it is possible to manipulate the crystallization process via applying external electric fields, which also serve as the poling process (*Materials Today* 24, 17–25 (2019), *J Am Chem Soc* 131, 2588–2596 (2009)). The in-situ electric field in the crystal growth process induces the domain alignment of β -glycine nanocrystals, suggesting the net polarization direction [020] is parallel to the electric field (Fig. 1e). The partially wet particles with incomplete evaporation deposited on the substrate are essential for further dense film formation in electro-spray deposition techniques (*Anal Chem* 77, 3189–3197 (2005), *ACS Appl. Polym. Mater.* 2023, 5, 3, 1797–1809, *Journal of Materials Chemistry*, 1996, 6(5): 765-771). During the synthesis of β -glycine nanocrystalline films, the crystallization is nearly completed before depositing on the substrate, while numerous nanocrystals still carry a thin water shell and further cluster together to form compact films (Fig. 1f). Notably, the dominant OOP orientation of the nanograins remains in the strongest polar direction [020].

Fig. 1 Fabrication of piezoelectric β -glycine nanocrystalline films and the active self-assembly mechanism via synergistic nanoconfinement and *in-situ* poling. **a**, Schematic of the bio-organic films printer and the synthesis of β -glycine nanocrystalline films. **b**, Schematic of the nano-micro droplet of glycine solution and the crystallization process. **c**, Illustration of the free energy (ΔG_{cryst}) profile of a growing crystal nucleus as a function of crystal radius, r . The energy profile results from the sum of the favorable volume free energy, ΔG_V , and the surface free energy ΔG_S . The profile passes through a maximum value of ΔG_{cryst} at the critical radius, r_c . **d**, Illustration of the size-dependent free energy profiles for two competing nuclei corresponding to α -glycine and β -glycine. **e**, Schematic of orientation alignment of glycine molecules during homogeneous nucleation. Molecular dipoles in β -glycine sum to a spontaneous polarization (red arrow P) along the 2-axis parallel to the electric field (black arrow E), which contributes to the longitudinal 22 piezoelectric coefficient. Molecules are displayed in the CPK coloring, including carbon (cyan), hydrogen (white), oxygen (red), and nitrogen (navy blue) atoms. The green arrow represents the dipole orientation of individual glycine molecule. **f**, Schematic of the film formation process showing the compact nanograins with uniform and consistent polarization orientation (red spot and red arrow in nanograins). The top two images are the surface view of films, and the bottom image is the cross-sectional view. **g**, Photographs of a film on a 4-inch silicon wafer (left) and film on a flexible gold-coated polyethylene terephthalate (PET) substrate (right).

4. The estimation of piezoelectric coefficients by the authors is also doubtful. The authors declared the d_{33} value of 11.2 pm V⁻¹ and wrote that “this value is consistent with the measured value (15 pm V⁻¹) of β -glycine microcrystals in previous reports³⁵ but larger than the calculated value (4.7 pm V⁻¹)²¹ since the giant shear piezoelectricity of β -glycine (178 pm V⁻¹) also contributes to the OOP piezoresponses^{36,37}”. This one sentence contains some errors.

a. First, the presented values of d_{33} and a “giant shear piezoelectricity” for β -glycine calculated

in Ref. 21 are incorrect. The correct values from that paper are 5.7 and 195 pm V⁻¹, respectively.

Response: Thanks for the referee’s careful examination. We have corrected the calculated piezoelectric coefficients of β -glycine crystals in the revised manuscript as below.

Results

The OOP amplitude increases linearly as a function of the applied AC voltage, and the slope yields the effective piezoelectric coefficient around 11.2 pm V⁻¹ (red curve in Fig. 4a). This value is consistent with the measured value (15 pm V⁻¹) of β -glycine microcrystals in previous reports (*Advanced Materials*, 2020, 32(46): 2002873). It is larger than the calculated value (5.7 pm V⁻¹) (*Nat Mater* 17, 180–186 (2018)) since the giant shear piezoelectricity of β -glycine (195 pm V⁻¹) contributes noticeably to the effective longitudinal piezoresponses when the OOP orientation is not strictly along the longitudinal direction (2 direction) of crystallography (*Cryst Growth Des* 18, 4844–4848 (2018); *J Appl Phys* 95, 1958–1962 (2004)).

b. Second, the “giant shear piezoelectricity”, obviously related to the d₁₆ piezoelectric coefficient, cannot contribute to the out-of-plane (OOP) signal detected by the authors. Following equation (1) from Ref. 37, the OOP response includes only coefficients d₃₁, d₁₅, and d₃₃. The coefficient d₁₆ cannot contribute to the vertical deformation of the crystal under an electric field.

Response: We thank the referee for this constructive comment. According to the Ref. 37 (*J Appl Phys* 95, 1958–1962 (2004)), the shear piezoelectric coefficient d₁₅ could contribute to the out-of-plane piezoresponses, with the OOP direction being along the 3 direction. For the piezoelectric tensor of β -glycine crystals, there are no d₃₃ and d₁₅, and the OOP indicates the 2 direction. However, through the matrix transformation whereby the 2 and 3 directions are exchanged as shown in the following figure, obtain a transformed piezoelectric tensor matrix that features d₁₅ (originally d₁₆) and d₃₃ (originally d₂₂). Based on this transformation and the equation in Ref. (*J Appl Phys* 95, 1958–1962 (2004)), we can infer that the giant shear piezoelectricity of β -glycine is capable of contributing to the OOP piezoresponses of the films.

$$\begin{pmatrix} 0 & 0 & 0 & d14 & 0 & d16 \\ d21 & d22 & d23 & 0 & d25 & d26 \\ 0 & 0 & 0 & d34 & 0 & d36 \end{pmatrix}$$

axis 2 \rightleftharpoons axis 3 axis 5 \rightleftharpoons axis 6

$$\begin{pmatrix} 0 & 0 & 0 & d14 & d15(d16) & 0 \\ 0 & 0 & 0 & d24(d34) & d25(d36) & 0 \\ d31(d21) & d32(d23) & d33(d22) & 0 & d35(d26) & d36(d25) \end{pmatrix}$$

c. Since the electromechanical response in PFM measurements is highly influenced by electrostatic interactions of non-piezoelectric origin, crystals with well-known piezoelectric tensors are typically used to calibrate the measured values. Lithium niobate is one of the most studied piezoelectric crystals with the table value of $d_{33} = 6.2 \text{ pm V}^{-1}$ [Opt. Mater. 23, 403-408 (2003), doi: 10.1016/S0925-3467(02)00328-2], whereas the authors measured the raw value 17.2 pm V^{-1} . Therefore, a correction factor of $6.2/17.2 = 0.36$ should be used to get the correct values of d_{33} . Taking into account this factor, for β -glycine, the $d_{33} = 11.2 \times 0.36 = 4.03 \text{ pm V}^{-1}$, which is a more reliable value comparable with that predicted in Ref. 21.

Response: Thanks for the comments. We agree with the referee's statement that it is important to calibrate the measured values by PFM using standard piezoelectric materials. We previously conducted the PFM measurements on the PPLN and collagen films, and obtained piezoelectric coefficients of 17.2 pm/V and 1.6 pm/V , respectively. It is worth noting that for PPLN, there is no universally agreed-upon standard value. In a recent study (*Nat Commun* 8, 1113 (2017)), A. Gomez et al. performed a direct quantitative analysis of the piezoelectric coefficient of PPLN, BFO, and PZT by applying a force and recording the generated current. They qualified the d_{33} for PPLN ($14 \pm 3 \text{ pC/N}$) and BFO ($43 \pm 6 \text{ pC/N}$) in agreement with the values reported in the previous literature ($6\text{--}16 \text{ pC/N}$, *Appl. Phys. A. Solids. Surf.* 37, 191–203 (1985)). Moreover, Seung-Wuk Lee et al. conducted the same PFM measurements on PPLN and obtained a d_{33} of 17 pm/V (*ACS Nano* 2018, 12, 8, 8138–8144). All these reported values of PPLN measured by various techniques are consistent with the presented value in our study.

To further verify the effectiveness of our PFM measurements, we have performed the same measurements on the commercial PVDF film with a thickness of $28 \text{ }\mu\text{m}$. The measured d_{33} is 25.2 pm/V , which is in good agreement with the bulk measurement value. In addition, we also

performed the ferroelectric hysteresis loop measurements on the β -glycine nanocrystalline films using the DART-SS-PFM (See the detailed response in Question 7). From the saturated piezoelectric response of the amplitude hysteresis loop measured during DC off state, we determined the piezoelectric coefficient d_{33} of approximately 13.3 pm/V. Furthermore, we have conducted the d_{33} meter measurements at the bulk scale obtained a piezoelectric coefficient of about 11 pm V⁻¹. The standard samples of PVDF and PZT as well as the non-piezoelectric paper were also tested for verifying the measurement reliability.

Overall, these additional measurements strengthen the validity of the presented piezoelectric coefficients of our films and provide further support for the effectiveness of our PFM measurements. We have added the measured results of PVDF, ferroelectric hysteresis loop, and d_{33} meter in the revised manuscript as below. Thank you for your careful consideration.

Results

To justify the measured piezoelectric coefficient using PFM, the same measurements on the periodically poled lithium niobate (PPLN), commercial polyvinylidene difluoride (PVDF) film, and collagen films are also performed for comparison (Fig. s15, Fig. s16, and Fig. s17). We then plot their piezoresponses under AC voltages that show the piezoelectric coefficients of about 17.2 pm V⁻¹ (PPLN), 25.2 pm V⁻¹ (PVDF), and 1.6 pm V⁻¹ (collagen) (grey curve, green curve, and blue curve in Fig. 4a), in good agreement with the value measured by quasi-static d_{33} meter or reported in the previous literatures (*Nat Commun* 8, 1113 (2017)). Furthermore, we perform the ferroelectric hysteresis loop measurements and show that the β -glycine nanocrystalline films are also ferroelectric (Fig. 4b, Fig. 4c) (*Ferroelectrics*, 2015, 475(1): 107-126; *Materials* 2019, 12(8), 1223)). We determine the piezoelectric coefficient of approximately 13.3 pm V⁻¹ based on the saturated piezoelectric response of the amplitude hysteresis loop measured during the DC off state (Fig. 4c). This value is consistent with the linear fitting one, manifesting the effectiveness of the measured piezoelectric coefficients. Additionally, we also conduct the d_{33} meter measurements and obtain a piezoelectric coefficient of about 11 pm V⁻¹, demonstrating a consistent piezoelectric property of the films at both the bulk scale and nanoscale (Fig. s18).

Fig. 4 Measurement of effective piezoelectric coefficients and ferroelectric hysteresis loops. **a**, Linear dependence of PFM amplitude on the applied AC voltage. Error bar denotes the standard deviation. **b**, Ferroelectric amplitude and phase hysteresis loops with the DC filed on. **c**, Ferroelectric amplitude and phase hysteresis loops with the DC filed off. **d**, Piezoelectric coefficient of as-prepared β -glycine nanocrystalline films compared with other bio-organic piezoelectric materials. Measured values of the piezoelectric coefficient are listed at the end of each bar.

Figure S16. PFM measurements of commercial PVDF thin film (thickness: 28 μm). **a**, AFM surface topography mapping. **b**, PFM OOP amplitude mapping. **c**, OOP phase mapping.

Figure S18. Macroscopic piezoelectricity of the β -glycine nanocrystalline films on aluminum foil (a), commercial PVDF film (b), PZT for calibration (c), and non-piezoelectric printer paper (d) measured by a commercial d_{33} meter.

d. The recalculation of d_{33} considering the correction factor will also lead to the reduction of the piezoelectric voltage coefficient g_{33} , which is will be $90 \times 10^{-3} \text{ VmN}^{-1}$ (and it is not the record value anymore).

Response: Thanks for the comments. We appreciate the referee's concerns regarding our claimed d_{33} value of 11.2 pm/V , because it is larger than the predicted value reported in a previous study (*Nat. Mater.* 17, 180–186 (2018)). Therefore, we would like to provide following additional context to address these concerns.

In the aforementioned literature, Sarah Guerin et al. calculated the longitudinal piezoelectric coefficient $d_{22} = 5.7$ pm/V using the DFT and measured the $d_{22} = 4.7$ pm/V based on resonance testing. Actually, in a subsequent study (*Advanced Materials*, 2020, 32(46): 2002873), the same group reported the longitudinal piezoelectric coefficients d_{33} (d_{22}) = 15 pm/V for β -glycine crystals and $d_{33} = 5.9$ pm/V for γ -glycine crystals using the SHG measurements. For the resonance testing, they cut and electrode the crystal precisely to isolate the longitudinal response only. However, for SHG, they measured what is called an “effective” longitudinal response. This means that the piezoresponse is primarily longitudinal, but there will be contributions from other piezoelectric tensor components also, particularly shear. For piezoelectric crystals like β -glycine that have large shear components, this will have a more noticeable contribution. In Ref. (*Cryst. Growth Des.* 2018, 18, 9, 4844–4848), the measurable longitudinal piezoelectricity of amino acid films with an orthorhombic crystal structure that precludes the existence of longitudinal piezoelectric coefficients further verifies the importance of shear piezoelectricity on the OOP piezoresponses of the oriented polycrystalline films. In addition, another study also reported an OOP piezoelectric coefficient $d_{\text{eff}} = 12.5$ pm/V of the β -glycine nanocrystal confined in polymer fiber (*Cryst. Growth Des.* 2011, 11, 10, 4288–4291).

Based on the above analysis and experiments, we are confident that our reported d_{33} value of 11.2 pm V^{-1} and derived g_{33} value of 252×10^{-3} Vm N^{-1} are reasonable and effective.

5. However, the main doubts are related to energy harvesting measurements. Though β -glycine indeed possesses high piezoelectric properties, it is a very fragile material demonstrating a tendency to transition into more stable phases not only under heating but mechanical stress as well [*J. Therm. Anal. Calorimetry* 73, 419–428 (2003), doi: 10.1023/A:1025457524874]. While measuring response under stress, the authors applied very high forces comparable with the action of a hummer drill (forces up to 9 N with frequency up to 40 Hz), and the film survived 24000 cycles without any degradation! Moreover, the film demonstrated unconceivable values of short-circuit current and output power of 3-6 orders of magnitude higher than in any other material (Table S3)! That looks fantastic and very unlikely! I am afraid it could be an error in the calculations and/or in the measurements and should be double/ or triple-checked. Did the authors see any mechanical damage after cycling?

Response: We thank the referee for the constructive comments and suggestions.

Regarding mechanical stability, in the literature given by the referee (*J. Therm. Anal. Calorimetry* 73, 419–428 (2003)), E. V. Boldyreva observed that the dry samples of the β -

glycine crystals were stable not only with respect to grinding, but also with respect to a prolonged mechanical treatment in much more powerful mechanical activators, than a mortar (Fritsch-5, AGO-2). In addition, Perlovich et al. also did not observe any $\beta \rightarrow \alpha$ transformation to be induced by grinding while they found the $\gamma \rightarrow \alpha$ transformation (*J. Therm. Anal. Cal.*, 66 (2001) 699.). It can be inferred that the stability of the β -glycine crystal is not sensitive to external mechanical force. Additionally, when the crystal size is constrained to nanometer-scale dimensions, an otherwise metastable form of a crystalline substance, β -glycine, actually becomes the stable form according to the Ostwald step rule (*J Am Chem Soc* **131**, 2588–2596 (2009), *Cryst Growth Des* 8, 3368–3375 (2008).).

As per the referee's suggestions, we fabricated five more samples and performed tests. We conducted the additional XRD and SEM measurements on the films after subjecting them to 24000 cycles of durability tests to verify the mechanical stability of the β -glycine nanocrystalline films. The testing results confirm that the nanocrystalline films retain their structural integrity and remain resistant to mechanical stress even after prolonged durability testing. The additional results and description have been included to the revised manuscript and supplementary information as below.

Results

After 24,000 compressing cycles, the open-circuit voltage output remains almost unchanged (Fig. 4e). The XRD and SEM results confirm neither phase transition nor mechanical damage

after cycling (Fig. 6h, Fig s26), manifesting the high durability and reliability of the films.

Fig. 6 Piezoelectric electrical performance and mechanical stability of β -glycine nanocrystalline films. **a**, Schematic of the piezoelectric device measurement set-up. The bottom right image is the photography of a real device. Scale bar: 1.5 cm. **b**, Open-circuit voltage and **c**, short-circuit current responses of the piezoelectric device of as-prepared films under 1.5 MPa compressive pressure with 30 Hz frequency. **d**, Linear dependence of the maximum open-circuit voltage output of the piezoelectric devices on the increased external forces. **e**, Dependence of the power output of the piezoelectric device on the resistance of the external load under 1.0 MPa compressive pressure. **f**, The charging curves of the 4.7 μF and 10 μF capacitor. **g**, Voltage output signals during 24,000 cycles under 1.5 MPa compressive pressure. **h**, XRD spectra of the films before and after durability tests of 24,000 compressing cycles. The standard XRD spectra of β -glycine are shown at the bottom of the figure. **i**, Optical image showing 3 LEDs lit up by an individual piezoelectric device through finger tapping with a frequency of 0.5 to 1 Hz.

Figure S26. Surface topography SEM image of the films after durability tests of 24,000 compressing cycles.

As for the high output performance, the β -glycine nanocrystalline films generate a power density of up to $3.61 \mu\text{W cm}^{-2}$ connecting to a load resistor of $2.6 \text{ M}\Omega$, which is one to three orders of magnitude higher than other piezoelectric biomaterials-based power generators. We attributed it to two key factors: the record-high piezoelectric coefficient and property uniformity of the films. To the best of our knowledge, this is the first time that both optimal polar orientation in the OOP direction and the compact, dense film structure have been achieved simultaneously in the single-component biopiezoelectric films.

In the previously reported piezoelectric biomaterial films, their polarization direction is either antiparallel in-plane (IP) or at a certain angle with the out-of-plane direction, which greatly weakens their piezoelectricity. For example, the antiparallel in-plane polarization of virus-based nanogenerators canceled the piezoelectricity of each other, severely limiting the output performance (*Nature Nanotech* 7, 351–356 (2012)). Although later studies have demonstrated the vertically aligned phage with OOP piezoelectricity using the AAO and PDMS template, the output still suffered from the antiparallel polarization and the retained template in the nanogenerator (*Energy Environ. Sci.*, 2015,8, 3198-3203; *Nano Lett.* 2019, 19, 4, 2661–2667). Similarly, γ -glycine or DL-alanine crystals packed between two electrodes were loosely

distributed with disordered polarization orientation, resulting in a significant reduction in the piezoelectric output (*Nat. Mater.* 17, 180–186 (2018); *Phys. Rev. Lett.* 122, 047701). Even though γ -glycine-PVA films were demonstrated with uniform and decent piezoelectricity, they only exhibit a portion of the intrinsic strongest piezoelectricity due to the polarization orientation [001] being not parallel to out-of-plane [101] (*Science*, 2021, 373(6552): 337-342). Additionally, the non-piezoelectric PVA layer reduces the short-circuit current and output power.

In our study, we performed the polarized Raman, XRD, and PFM measurements to confirm the strongest piezoelectric orientation of β -glycine (020) is indeed in the OOP direction, which is analogous to the textured inorganic ceramic with enhanced piezoelectricity. The topography and cross-sectional SEM images also showed the compact, dense, and uniform structure of the films. The signs of voltage output in agreement with the polarization direction, the reversed outputs by reversing the connection, and the linearity with force all verified that the measured electrical signal is truly from the piezoelectricity. An individual device also lit up three LEDs, indicating its reliable power output capability.

To further demonstrate the effectiveness of the piezoelectric device measurements, we have conducted the control device experiments of non-piezoelectric Kapton PI film and commercial PVDF film for comparison, which are included in the revised manuscript and supplementary information as below. The non-piezoelectric Kapton PI film-based device hardly exhibited evident output signal compared with the β -glycine films-based devices. On the contrast, the commercial PVDF film-based device demonstrated both a higher output short-circuit current (9 μA) and higher power density (16.11 $\mu\text{W cm}^{-2}$) than those of the β -glycine films-based devices (4 μA and 3.61 $\mu\text{W cm}^{-2}$). This is reasonable because the PVDF film holds a higher d_{33} and a comparable g_{33} value with β -glycine films.

Results

The β -glycine nanocrystalline film produces a maximum open-circuit voltage of 14.5 V (Fig. 6b, control devices are shown in Fig. s21 and Fig. s22), and a maximum short-circuit current of 4 μA (Fig. 6c), which are an order of magnitude larger than most reported piezoelectric biomaterials (Table s3).

Methods

Piezoelectric device fabrication: By replacing the film with a non-piezoelectric Kapton PI tape or commercial PVDF films, we designed the control devices with the structural assembly and measurement parameters consistent with the β -glycine film-based piezoelectric devices.

Figure S21. Comparison of the open-circuit voltage measurements of the piezoelectric device of β -glycine nanocrystalline films and control device of non-piezoelectric Kapton PI film.

Figure S22. PVDF film-based piezoelectric device measurements. a, The measured open-circuit voltage. b, The measured short-circuit current. c, Dependence of the power output of the piezoelectric device on the resistance of the external load.

6. A little correction for the thermal measurements. The authors wrote (line 204–205): “we confirm the decomposition temperature of β -glycine nanocrystalline films is 255 °C”. However,

this is incorrect. The nanocrystalline film stops existing at the moment of melting at 192 °C. At 255 °C, the glycine molecules evaporation probably takes place.

Response: We thank the referee for the careful review and professional comments. We have corrected the description in the revised manuscript as suggested.

Results

Combined with the thermal gravimetric analysis (TGA) results, we confirm the decomposition temperature of the glycine molecules is 255 °C.

This suggests that the disappearance of β -glycine nanocrystalline films at 192 °C is due to melting, where the melting point depression effect is expected from the nanoscale crystal size (supplementary text).

7. Is there any sign of ferroelectricity in these films? What is the phase transition temperature? Is there any sign of Curie-Weiss behavior in the dielectric constant? These complementary data may give some clue for the unusually high energy harvesting figures of merit.

Response: Thanks for the referee's constructive comments and suggestions. The β -glycine crystals have been reported with ferroelectricity demonstrated by both computational simulations and experiments (*Ferroelectrics*, 2015, 475(1): 107-126; *J. Appl. Phys.* 118, 072008 (2015); *Materials* 2019, 12(8), 1223). To verify the ferroelectricity in our films as suggested, we have conducted the ferroelectric switching measurements using the SSPFM. To minimize the effects of electrostatic interactions, the piezoresponse is measured during "off" state at each step, and the phase-voltage hysteresis loop is evident. The reversal in the piezoresponse phase occurs when a coercive voltage is exceeded, approximately 3.3 V on the positive side and -5 V on the negative side. The phase contrast is approximately 180°, clearly indicating polarization switching. The amplitude-voltage butterfly loops are also observed which saturates at a relatively high voltage, suggesting that the response is piezoelectric instead of electrostatic, and thus the phase reversal does signal polarization switching and ferroelectricity. This is also confirmed by the corresponding loops measured during "on" state, where the coercive voltage is substantially smaller, and the responses are higher than those measured during off state, due to strong contributions from electrostatic interactions. The differences between on and off states are evident, confirming the phase reversal observed during off state is indeed ferroelectric.

Results

Furthermore, we perform the ferroelectric hysteresis loop measurements and show that the β -

glycine nanocrystalline films are also ferroelectric (Fig. 4b, Fig. 4c) (*Ferroelectrics*, 2015, 475(1): 107-126; *Materials* 2019, 12(8), 1223)). We determine the piezoelectric coefficient of approximately 13.3 pm V^{-1} based on the saturated piezoelectric response of the amplitude hysteresis loop measured during the DC off state (Fig. 4c). This value is consistent with the linear fitting one, manifesting the effectiveness of the measured piezoelectric coefficients.

Fig. 4 Measurement of effective piezoelectric coefficients and ferroelectric hysteresis loops. **a**, Linear dependence of PFM amplitude on the applied AC voltage. Error bar denotes the standard deviation. **b**, Ferroelectric amplitude and phase hysteresis loops with the DC filed on. **c**, Ferroelectric amplitude and phase hysteresis loops with the DC filed off. **d**, Piezoelectric coefficient of as-prepared β -glycine nanocrystalline films compared with other bio-organic piezoelectric materials. Measured values of the piezoelectric coefficient are listed at the end of each bar.

For the phase transition temperature, we previously conducted the DSC, TGA, and in situ XRD measurements and confirmed the absence of a Curie transition prior to the melting temperature of $192 \text{ }^\circ\text{C}$. In response to the referee's suggestions, we have performed the in situ variable temperature Raman testing and dielectric-temperature measurements to further verify the statement and confirm whether the film exhibits Curie-Weiss behavior. The in-situ Raman spectra of the films exhibit only β phase prior to the melting temperature. The measured relative permittivity at different frequencies shows no evident anomaly before about $170 \text{ }^\circ\text{C}$ while a sharp change in the range of 170 to $190 \text{ }^\circ\text{C}$, indicating no phase transition in the β -glycine nanocrystalline films and the film's destruction after $170 \text{ }^\circ\text{C}$ due to the melting, which is in good agreement with the DSC and in situ XRD results. The subtle differences in temperature

results may be caused by the instability of the temperature testing module or electrical breakdown resulting from the destruction of film structure. Furthermore, no evident frequency dispersion is observed in temperature-dependent permittivity curve, indicating that there is no sign of Curie-Weiss behavior. The additional results and description have been added to the revised manuscript as below, which provides a more comprehensive understanding of the phase transition behavior of the β -glycine nanocrystalline films.

Results

To further verify whether the thermal anomaly at 192 °C is from phase transition or from melting, we conduct *in-situ* XRD measurements (Fig. 5b) and observe no phase other than the β phase throughout the heating process, which is also verified by the *in-situ* Raman measurements (Fig. 5c). The weak peak at 2θ of about 26.9 degree in the XRD pattern corresponds to the peak of the silicon substrate used in the test, and it persists after the decomposition of the glycine molecules. The peak position shifts in the XRD pattern and Raman spectra could be attributed to the thermal expansion in the crystal lattice with increasing temperature. The temperature dependency tests of relative permittivity at different frequencies also shows no evident anomaly before electrical breakdown of the films, indicating the absence of a Curie transition prior to the melting temperature (Fig. 5d). This suggests that the disappearance of β -glycine nanocrystalline films at 192 °C is due to melting, where the melting point depression effect is expected from the nanoscale crystal size (supplementary text).

Fig. 5 Thermostability and overview of piezoelectric properties. **a**, DSC (black curve) and TGA (blue curve) results of as-prepared films. The inset figure is the enlarged drawing of DSC curve between the temperature of 180 °C and 205 °C. **b**, The *in-situ* variable temperature XRD patterns of the films. **c**, The *in-situ* variable temperature Raman spectra. **d**, The temperature-dependent relative permittivity for selected frequencies. **e**, Dependence of d_{33} of β -glycine nanocrystalline films on temperature. **f**, PFM OOP amplitude mapping (left) and phase mapping (right) under 120 °C confirming the stable piezoelectric effect of the films under high temperature. The applied AC voltage is 2 V. Scale bar: 400 nm. **g**, Comparison of longitudinal piezoelectric voltage coefficient g_{33} of most actively studied piezoelectric material systems with this work as a function of Curie temperature T_C .

8. The section on DL-threonine films and crystals is totally abundant. It takes only one paragraph and does not report any essential results. This part should be completely deleted.

Response: Thank the referee for the constructive comments. In this work, we aimed not only to present the high-performance piezoelectric β -glycine nanocrystalline films with enhanced thermostability, but also to propose a generalizable route to actively self-assemble the

biomolecular crystals with optimally aligned polarization orientation in addition to uniform and compact polycrystalline structure. Our strategy is expected to be applied to various biomaterials and other piezoelectric materials, such as molecular or organic-inorganic piezoelectric materials. Furthermore, it should be scalable to create films with variable dimensions, programmable structures, and diverse material forms such as flexible composites. However, we acknowledge that the section on DL-threonine films was not thoroughly investigated and analyzed. Therefore, we decided to remove this section from the revised manuscript, as suggested by the reviewer.

To summarize, the paper by Z. Zhang et al. is not ready for publication in its current form and needs additional proof and reconsideration of the experimental results. Nevertheless, the suggested technique is promising and may be of great interest to readers of a more specialized journal.

Response: We are pleased to see that the referee found our presented technique promising and may be of great interest to a large community of readers. We actually thank you very much for your many critical comments that take us months to add more experiments. Thank Editor to find such a great scholar who knows this topic very well. As identified by Reviewer 1, there are very limited scholars in the world who fabricate β -glycine crystals and stabilize them. Many known physics are waiting to be found in this nanoscale phase transformation. Upon receiving your comments, we are surprised by your profound knowledge on this topic. Your excellent comments push us to go back and check every corner of this two-year project and reexamine our hypothesis. Thank you very much. Our group has intensively worked in the past three months address all the concerns raised by the referee, including the piezoelectric coefficient, ferroelectricity, and mechanical stability of the films. We hope that the referee is now happy with our revised manuscript. We will keep working on glycine and looking forward to unveiling new piezoelectric/ferroelectric and nucleation phenomenon.

Reviewer #3 (Remarks to the Author):

This work reports a synthesis of β -glycine film by the electrohydrodynamic spray method. Electric field applied during the spray was claimed being able to lead the formation of β -glycine and align the piezoelectric polarization. The as-synthesized film showed a strong piezoelectricity, and retained it up to 192 °C. The results are interesting and might be impactful as a new piezoelectric bio-crystal thin film. However, there are some key factors that are defective, which undermine the significance of this work.

1. The most important issue is the phase and associated good thermostability. It is known that the β -glycine is not stable at high temperature, which is also highlighted by the authors as the main novelty. However, this important advancement is not explained. Why this polycrystalline film can stabilize the beta phase? Either thermodynamically or kinetically, it does not make sense simply by forming such a film, unless the phase itself is stable. Authors need to carefully analyze the XRD spectra, as there were only a few peaks shown. The strong peak may also come from other phases such as the gamma phase. As temperature increases, the strong peak also shifted to lower angle. Why? In order to support that it is the true beta phase property, authors need to carefully confirm the phase with more detailed diffraction analysis on peak positions for different phases, and provide a reasonable explanation why this film or approach can substantially improve the beta-phase stability. The explanation should also be supported by control experiments to validate the hypothesis.

Response: We appreciate that the referee thought our work interesting and acknowledged the potential impact of developing piezoelectric biomaterial films. There are several points embedded in this comment, for which we would like to address one-by-one:

- **Thermostability of β -glycine nanocrystalline films:** As suggested by the referee, the β -glycine crystal phase is known to be metastable and readily transforms into the more stable α phase upon heating (*J. Therm. Anal. Calorimetry* 73, 419–428 (2003)). The crystal structure may depend on both kinetics and thermodynamics, as only one phase can correspond to the global minimum of free energy at a specific pressure and temperature. In this case, at the early stages of crystal growth, the consolidation of molecules into prenucleation aggregates and crystal nuclei of a critical size, are important for determining the growth trajectory of the crystal and the characteristics of the initially formed polymorph. This recalls the Ostwald step rule, which assumes that the least stable polymorphs crystallize first during the crystallization process. Ostwald step rule can be seen as a requirement for polymorphic systems - if the most stable form

crystallizes first, polymorphism is unlikely to be observed. Nevertheless, the step rule raises a question as to whether polymorphs formed at the early stages of crystallization are favored due to their small size. This could possibly be explained using classical nucleation theory:

The unique properties of nanometer-scale crystals, confined by the nanometer-scale containers, can be attributed to increasing the surface area-to-volume ratio as the crystal size is reduced (*Israel Journal of Chemistry*, 2017, 57(1-2): 82-92.). According to the classical nucleation theory, small clusters of molecules form in the early stage, and then grow into a nucleus. The free energy of the nucleus is equal to the sum of the volume free energy change ΔG_V and adverse surface free energy ΔG_S (Fig. 1c). The volume free energy is always negative and stable, which can be attributed to the intermolecular bonding energy, while the surface free energy is positive and unstable, which corresponds to the interface formation between the nucleus and surroundings (*Crystallization*. Elsevier, 2001.). With regard to the typical spherical nucleus, the free energy along the crystallization path can be described as follows:

$$\Delta G_{cryst} = \Delta G_V + \Delta G_S = \frac{4}{3}\pi r^3 \Delta g + 4\pi r^2 \sigma \quad (1)$$

in which r is the radius of the spherical nucleus, Δg denotes the free energy gap between the nucleated phase and the nucleating phase for a unit volume, and σ is the surface tension of the interface and represents the surface free energy in each unit area. According to the Eq. (1), it is obvious that ΔG_{cryst} is strongly related to the crystal size. The maximum value of ΔG_{cryst} can be obtained by derivating it with regard to r , corresponding to the activation energy of nucleation ΔG_c at the critical radius r_c . It is critical to surmount the energy barrier for spontaneous nucleation. Dissolution occurs when nuclei is smaller than r_c , while crystallization occurs when nuclei is larger than r_c . The critical size of organic nuclei typically ranges from a few nanometers to tens of nanometers (*Polymorphism in Molecular Crystals 2e*. (2020)).

Because of the distinct crystal structures of polymorphs, their specific surface energies, volume free energies, and crystal morphologies should also be different. It can be reasonably inferred that each polymorph shall possess different values of ΔG_{cryst} and r_c . Here is where thermodynamics and kinetics collide. At the critical size, the difference in kinetic barriers of the two polymorphs is equivalent to the difference in their thermodynamic stability. When considering the trajectory of the nucleation of different forms, it is expected to have different critical sizes and different corresponding

nucleation barriers. Fig. 1d shows free energy profiles where α phase glycine crystals are more stable in bulk sizes, whereas metastable β phase glycine crystals are more stable at the critical size and slightly beyond. With dimensions slightly beyond the critical size, β phase is the thermodynamically preferred phase corresponding to the lowest free energy and lower kinetic barrier compared with the other phases. This ranking can persist beyond the critical size but reverse to the stability ranking of the bulk as the size increases. It demonstrates that an otherwise metastable form of a crystalline substance, β -glycine, actually becomes the stable form when the crystal size is constrained to nanometer-scale dimensions.

More than a decade ago, the effect of pore size on crystal polymorphism was demonstrated for anthranilic acid (AA), a compound with three known polymorphs. When AA was crystallized in mesoporous silica with average pore diameters of 7.5, 24, and 55 nm, form III was obtained in 55 nm pores and on the surfaces of nonporous glass beads, while a mixture of forms II and III was obtained in 24 nm pores, and form II was exclusively obtained in 7.5 nm pores (*J. Am. Chem. Soc.* 2004, 126, 3382–3383.). Interestingly, form III is the thermodynamically stable phase in the bulk, yet form II persisted in the pores indefinitely. Since these seminal findings on size-dependent polymorphism, numerous reports have revealed the effects of nanoscale confinement on polymorph stability rankings and phase behavior, for instance, with the crystallization of α,ω -alkanedicarboxylic acids such as glutaric acid, pimelic acid, and suberic acid, and coumarin (1,2-benzopyrone), in controlled pore glass (CPG) and nanoporous poly(cyclohexylethylene) polymer monoliths (*Cryst. Growth Des.* 2009, 9, 4766–4777.).

Moreover, β -glycine nanocrystals embedded in CPG and anodic aluminum oxide (AAO) were also demonstrated to be indefinitely stable in ambient air (*J. Am. Chem. Soc.* 2005, 127, 14982–14983; *Cryst. Growth Des.* 2008, 8, 3368–3375; *J. Am. Chem. Soc.* 2013, 135, 2144–2147). In addition, embedded β -glycine nanocrystals also were more stable against phase transitions at elevated temperatures compared with bulk β -glycine. In CPG, the β form persisted well above the bulk β - α phase transition temperature (67 °C), melting near 180 °C. These observations along with our finding that the media free β -glycine nanocrystals produced by electrospray method and the corresponding nanocrystalline films exhibit excellent thermostability before melting (192 °C), collectively support the Ostwald step rule, which posits that metastable forms in bulk are typically, the more stable forms when confined in nanometer-scale dimensions.

Following the referee's suggestions, we have detailly discussed and analyzed the origin of favorable kinetics formation and thermostability of β -glycine nanocrystalline films in the revised manuscript as follows.

Results

With the rapid water evaporation and the increasingly large surface-area-to-volume ratio of the nano-micro droplets, the glycine nucleus is formed in the β phase through the nanoconfinement effect. Whereas the α polymorph forms most readily when glycine is crystallized from aqueous solutions, the metastable β polymorph is demonstrated to be favored in nanoscopic pores or micrometer-scale patterned substrates (*J Am Chem Soc* 135, 2144–2147 (2013); *J Am Chem Soc* **127**, 14982–14983 (2005)). This is supported by the Ostwald step rule that the least stable polymorph crystallizes first at the early stages of the crystallization because of their small size. It can be explained using the classical nucleation theory (*Acc Chem Res* **45**, 414–423 (2012)). The free energy of the nucleus is equal to the sum of the volume free energy change ΔG_V and adverse surface free energy ΔG_S (Fig. 1c). With regard to the typical spherical nucleus, the free energy along the crystallization path can be described as follows

$$\Delta G_{cryst} = \Delta G_V + \Delta G_S = \frac{4}{3}\pi r^3 \Delta g + 4\pi r^2 \sigma \quad (1)$$

in which r is the radius of the spherical nucleus, Δg denotes the free energy gap between the nucleated phase and the nucleating phase for a unit volume, and σ is the surface tension of the interface and represents the surface free energy in each unit area. From the Eq. (1), it is obvious that ΔG_{cryst} is strongly related to the crystal size. The maximum value of ΔG_{cryst} can be obtained by derivating it with regard to r ; corresponding to the activation energy of nucleation ΔG_c at the critical radius r_c . It is critical to surmount the energy barrier for spontaneous nucleation. Because of the distinct crystal structures of polymorphs, their specific surface energies, volume free energies, and crystal morphologies should also be different. It can be reasonably inferred that each polymorph shall possess different values of ΔG_{cryst} and r_c . Here is where thermodynamics and kinetics collide. At the critical size, the difference in kinetic barriers of the two polymorphs is equivalent to the difference in their thermodynamic stability. When considering the trajectory of the nucleation of different forms, it is expected to have different critical sizes and different corresponding nucleation barriers. Fig. 1d shows free energy profiles where α phase glycine crystals are more stable in bulk sizes, whereas metastable β phase glycine crystals are more stable at the critical size and slightly beyond. With dimensions slightly beyond the critical size, β phase is

the thermodynamically preferred phase corresponding to the lowest free energy and lower kinetic barrier compared with the other phases.

Fig. 1 Fabrication of piezoelectric β -glycine nanocrystalline films and the active self-assembly mechanism via synergistic nanoconfinement and *in-situ* poling. **a**, Schematic of the bio-organic films printer and the synthesis of β -glycine nanocrystalline films. **b**, Schematic of the nano-micro droplet of glycine solution and the crystallization process. **c**, Illustration of the free energy (ΔG_{cryst}) profile of a growing crystal nucleus as a function of crystal radius, r . The energy profile results from the sum of the favorable volume free energy, ΔG_V , and the surface free energy ΔG_S . The profile passes through a maximum value of ΔG_{cryst} at the critical radius, r_c . **d**, Illustration of the size-dependent free energy profiles for two competing nuclei corresponding to α -glycine and β -glycine. **e**, Schematic of orientation alignment of glycine molecules during homogeneous nucleation. Molecular dipoles in β -glycine sum to a spontaneous polarization (red arrow P) along the 2-axis parallel to the electric field (black arrow E), which contributes to the longitudinal 22 piezoelectric coefficient. Molecules are displayed in the CPK coloring, including carbon (cyan), hydrogen (white), oxygen (red), and nitrogen (navy blue) atoms. The green arrow represents the dipole orientation of individual glycine molecule. **f**, Schematic of the film formation process showing the compact nanograins with uniform and consistent polarization orientation (red spot and red arrow in nanograins). The top two images are the surface view of films, and the bottom image is the cross-sectional view. **g**, Photographs of a film on a 4-inch silicon wafer (left) and film on a flexible gold-coated polyethylene terephthalate (PET) substrate (right).

Results

The infinite thermostability of the films prior to melting temperature of 192 °C is good agreement with the observations in confined nanopores (*J Am Chem Soc* **131**, 2588–2596 (2009), *Cryst Growth Des* **8**, 3368–3375 (2008).), collectively supporting the

Ostwald step rule, which posits that an otherwise metastable form of a crystalline substance, β -glycine, actually becomes the stable form when the crystal size is constrained to nanometer-scale dimensions.

- **XRD spectra analysis:** In order to provide a clearer comparison of the XRD results and to eliminate the possibility of other phases, we have added the standard spectra of three phases of glycine crystals to each XRD pattern results in our revised manuscript. Through careful analysis, it has been determined that only the β phase exists in our films throughout the temperature dependence measurements with the strongest peak at 2θ of around 28.6 attributed to β phase (002) and the secondary peak at 2θ of about 23.6 attributed to β phase (110). There are no peaks attributed to the γ phase in the vicinity of this peak, confirming the absence of the γ phase. The weak peak at 2θ of about 26.9 corresponds to the peak of the silicon substrate used in the test, and it persists after the decomposition of the thin film molecules. Furthermore, there are no standard peaks associated with any phase of glycine crystals in the vicinity of this peak.

As for the peak shift, it is commonly present in in-situ variable temperature XRD testing. As the temperature increases, atoms in the crystal lattice vibrate more rapidly, leading to thermal expansion and changes in the interatomic distances and angles. This can cause a shift in the position of the diffraction peaks in the XRD pattern. In general, an increase in temperature can cause a shift in the peak position towards lower 2θ angles, indicating an increase in the lattice spacing (*Carbon Trends* 5 (2021): 100124). This can be attributed to the thermal expansion of the crystal lattice. Nevertheless, the instrumental effects such as thermal drift and sample displacement also have a significant impact on the in situ XRD measurement results.

- **Additional experiments:** We previously conducted the TGA, DSC, in situ XRD, and in situ PFM measurements to confirm the thermostability of the β -glycine nanocrystalline films. The DSC results reveal only two thermal anomalies at roughly 192 °C and 255 °C, and the in situ XRD results observe no phase other than β phase throughout the heating process.

Following the referee's suggestions, we here performed in-situ variable temperature Raman testing and dielectric-temperature measurements to further verify the thermostability of the films and confirm whether they exhibit phase-transition behavior.

The in-situ Raman spectra of the films exhibit only β phase prior to the melting temperature. The measured relative permittivity at different frequencies shows no evident anomaly before about 170 °C while a sharp change in the range of 170 to 190 °C, indicating no phase transition in the β -glycine nanocrystalline films and the film's destruction after 170 °C due to the melting, which is in good agreement with the DSC and in situ XRD results. The subtle differences in temperature results may be caused by the instability of the temperature testing module or electrical breakdown resulting from the destruction of film structure. The additional results and description have been added to the revised manuscript as below, which provides a more comprehensive understanding of the phase transition behavior of the β -glycine nanocrystalline films.

Results

To further verify whether the thermal anomaly at 192 °C is from phase transition or from melting, we conduct *in-situ* XRD measurements (Fig. 5b) and observe no phase other than the β phase throughout the heating process, which is also verified by the *in-situ* Raman measurements (Fig. 5c). The weak peak at 2θ of about 26.9 degree in the XRD pattern corresponds to the peak of the silicon substrate used in the test, and it persists after the decomposition of the glycine molecules. The peak position shifts in the XRD pattern and Raman spectra could be attributed to the thermal expansion in the crystal lattice with increasing temperature. The temperature dependency tests of relative permittivity at different frequencies also shows no evident anomaly before electrical breakdown of the films, indicating the absence of a Curie transition prior to the melting temperature (Fig. 5d). This suggests that the disappearance of β -glycine nanocrystalline films at 192 °C is due to melting, where the melting point depression effect is expected from the nanoscale crystal size (supplementary text).

Fig. 2 Morphology and structural characterization of β -glycine nanocrystalline films. **a**, Cross-sectional SEM image of an as-obtained film with the thickness of 4.5 μm . **b**, Surface topography SEM image showing the compact nanosized grains of the uniform and continuous films. **c**, Grain size distribution of glycine nanograins by analyzing over 400 grains. **d**, Raman spectrum of as-grown β -glycine nanocrystalline films. **e**, XRD spectra of as-prepared β -glycine nanocrystalline films (red curve), β -glycine nanocrystals obtained in the absence of electric field (green curve), β -glycine microcrystals prepared by electrohydrodynamic focusing deposition (blue curve), and α crystals formed by direct evaporation of glycine solution film (black curve). The standard XRD spectra of three phases of glycine are shown at the top of the figure. W/ denotes with, and W/O stands for without.

Fig. 5 Thermostability and overview of piezoelectric properties. **a**, DSC (black curve) and TGA (blue curve) results of as-prepared films. The inset figure is the enlarged drawing of DSC curve between the temperature of 180 °C and 205 °C. **b**, The *in-situ* variable temperature XRD patterns of the films. **c**, The *in-situ* variable temperature Raman spectra. **d**, The temperature-dependent relative permittivity for selected frequencies. **e**, Dependence of d_{33} of β -glycine nanocrystalline films on temperature. **f**, PFM OOP amplitude mapping (left) and phase mapping (right) under 120 °C confirming the stable piezoelectric effect of the films under high temperature. The applied AC voltage is 2 V. Scale bar: 400 nm. **g**, Comparison of longitudinal piezoelectric voltage coefficient g_{33} of most actively studied piezoelectric material systems with this work as a function of Curie temperature T_c .

2. The d_{33} was only measured by a PFM. This is not reliable and not adequate to claim the piezoelectric property of the large and thick films. PFM can easily be influenced by many surface features, such as large roughness or edge effect, and can show substantially different location-dependent results. Measuring only a very small area and use it to represent the entire film's property is not acceptable. Authors should provide bulk scale measurements to confirm the film's piezoelectric property.

Response: We thank the referee for the insightful comments and suggestions. In the past decades, PFM has been established as a powerful tool for nanoscale imaging, spectroscopy, and manipulation of ferroelectric and piezoelectric materials. There are also many wonderful frontier studies that have adopted the PFM to quantify the piezoelectric coefficients of piezoelectric 2D materials, thick molecular ferroelectric films, and biomaterials (*Science*, 2022, 376(6596): 973-978; *Science*, 2019, 363(6432): 1206-1210; *Nature Nanotech* 7, 351–356 (2012)). We agree with the referee’s statement that PFM has limitations on the piezoelectricity characterizations due to the effect of surface topography and measurement locations. Therefore, we have made a lot of efforts to lower the influence.

Firstly, we adopt a dual AC resonance tracking (DART) technique (*Phys. Rev. Lett.* 108, 078103), which measures the piezoresponse at two distinct frequencies across resonance, and use the amplitude difference at these two frequencies for feedback control. Measurements at two distinct frequencies allow us to solve for amplitude and phase at resonance, as well as the resonant frequency and quality factor. This makes it possible to determine the intrinsic piezoresponse mapping by correcting the resonance magnification using quality factor, which is able to largely address the impact of surface topography.

Secondly, we measured the piezoresponse mapping of multiple regions randomly selected from different samples under different AC drive voltages, and determined the piezoelectric coefficients by linear fitting, which can reduce the impact of location dependency.

Furthermore, we also performed the same PFM measurements on other standard samples such as PPLN, commercial PVDF film, and collagen films for calibration. The measured d_{33} of these samples is consistent with their bulk-scale value or previously reported value.

In addition, we also conducted single-point ferroelectric hysteresis loop measurements on our films using the DART-SSPFM technique to avoid the influence of surface features and scanning. The calculated d_{33} of about 13.3 pm/V is also in good agreement with the linear fitting value.

Following referee’s suggestion, we have conducted the d_{33} meter measurements at the bulk scale, and the results are shown below. The standard samples of PVDF and PZT as well as the non-piezoelectric paper were also tested for verifying the measurement reliability.

Results

To justify the measured piezoelectric coefficient using PFM, the same measurements on the periodically poled lithium niobate (PPLN), commercial polyvinylidene difluoride (PVDF) film, and collagen films are also performed for comparison (Fig. s15, Fig. s16, and Fig. s17). We

then plot their piezoresponses under AC voltages that show the piezoelectric coefficients of about 17.2 pm V^{-1} (PPLN), 25.2 pm V^{-1} (PVDF), and 1.6 pm V^{-1} (collagen) (grey curve, green curve, and blue curve in Fig. 4a), in good agreement with the value measured by quasi-static d_{33} meter or reported in the previous literatures (*Nat Commun* 8, 1113 (2017)). Furthermore, we perform the ferroelectric hysteresis loop measurements and show that the β -glycine nanocrystalline films are also ferroelectric (Fig. 4b, Fig. 4c) (*Ferroelectrics*, 2015, 475(1): 107-126; *Materials* 2019, 12(8), 1223)). We determine the piezoelectric coefficient of approximately 13.3 pm V^{-1} based on the saturated piezoelectric response of the amplitude hysteresis loop measured during the DC off state (Fig. 4c). This value is consistent with the linear fitting one, manifesting the effectiveness of the measured piezoelectric coefficients. Additionally, we also conduct the d_{33} meter measurements and obtain a piezoelectric coefficient of about 11 pm V^{-1} , demonstrating a consistent piezoelectric property of the films at both the bulk scale and nanoscale (Fig. s18).

Fig. 4 Measurement of effective piezoelectric coefficients and ferroelectric hysteresis loops. **a**, Linear dependence of PFM amplitude on the applied AC voltage. Error bar denotes the standard deviation. **b**, Ferroelectric amplitude and phase hysteresis loops with the DC filed on. **c**, Ferroelectric amplitude and phase hysteresis loops with the DC filed off. **d**, Piezoelectric coefficient of as-prepared β -glycine nanocrystalline films compared with other bio-organic piezoelectric materials. Measured values of the piezoelectric coefficient are listed at the end of each bar.

Figure S16. PFM measurements of commercial PVDF thin film (thickness: 28 μm). a, AFM surface topography mapping. b, PFM OOP amplitude mapping. c, OOP phase mapping.

Figure S18. Macroscopic piezoelectricity of the β -glycine nanocrystalline films on aluminum foil (a), commercial PVDF film (b), PZT for calibration (c), and non-piezoelectric printer paper (d) measured by a commercial d_{33} meter.

3. Piezoelectric and ferroelectric were both used in the paper randomly. It is not accurate. There is no ferroelectric property shown at all in the work. When talk about polarization alignment, piezoelectric refers to crystal orientation, and ferroelectric refers to ionic dipole orientation.

Response: We thank the referee for the professional comments. In the above responses, we have confirmed the ferroelectricity of the films. Nonetheless, given our work's primary focus on high piezoelectricity, we agree with the referee that we should make it clear on the statement of piezoelectric or ferroelectric. Consequently, we decided to use the term “piezoelectric” to describe the polarization alignment throughout the paper. The detailed correction in the revised manuscript is shown below:

Results

Page 1, Manuscript:

“...allows the electric field applied *in-situ* to align crystal grains...”

Page 4, Manuscript:

“...it is challenging to align the polarization of β -glycine microcrystals...”

Page 4, Manuscript:

“...indicating that the polarization of the as-prepared films is well aligned...”

Page 4, Manuscript:

“...exhibits both domains with opposite polarizations.”

Page 5, Manuscript:

“...be attributed to the excellent alignment of polarization...”

Page 7, Manuscript:

“...enables the large-scale out-of-plane (OOP) alignment of crystal grains in the strongest polarization direction by *in-situ* electric field...”

4. Aside from the defective materials synthesis and characterization approach and discussion, the application demonstration is routine and simple. No unique application that relies on the biomaterials nature is demonstrated, which make this work less attractive.

Response: We appreciate the referee’s constructive comments and suggestions. As a result of their endogenous biomaterials nature, our presented amino acid films can be used in biotechnological and medical fields, thus bridging the electronics and biological worlds. Due to limited experimental conditions, we have not yet found a suitable opportunity to conduct animal testing with our materials. In addition, the focus of this work is not only to present the

high-performance piezoelectric β -glycine nanocrystalline films with enhanced thermostability, but also to propose a generalizable route to actively self-assemble the biomolecular crystals with optimally aligned polarization orientation in addition to uniform and compact polycrystalline structure. However, we still appreciate the valuable suggestions of the reviewers and will consider exploring the unique biomedical application of our new piezoelectric biomaterials in future research.

Especially, we are encouraged by the latest article published in *Science* in April (*Science* 380,87–93 (2023)), in which the authors proposed a modified texturing process to fabricate textured PZT ceramics. Their new method is able to orient the inorganic crystal grains to improve the piezoelectric coefficients and thermostability. We believe that our work can also be impactful and embraced by a large community of researchers. We envision that many other researchers, whether specializing in novel materials, self-assembly, structure, biomedical engineering, or device fields, can draw inspiration from our work and unleash new innovative research.

REVIEWERS' COMMENTS

Reviewer #1 (Remarks to the Author):

In this revised version authors have considered referee comments in very high amount of details and addressed all major points of concerns and recommendations satisfactorily. Minor issue remains on the nature of crystallisation mechanism, which authors have argued to be of homogenous nature and minimal role of the solid substrate. I found their arguments convincing. I recommend publication of the revised manuscript with one additional comment: English corrections especially as regards to article use. There are numerous occasions of misuse (and often no use) of the definite article 'the'. Perhaps this could be further checked at the copy editor. Overall it is an excellent paper and a joy to read.

Reviewer #2 (Remarks to the Author):

Zhuomin Zhang and co-authors have made an enormous piece of work providing exhaustive explanations and additional data verifying and validating their impressive conclusions. New experiments successfully support the initial data. Although the suggested mechanism of the crystal growth at the nanoconfinement and under an electric field is still debatable and requires additional studies, the experimental methods and analyses are correct and do not raise doubts anymore. Indeed, the presented results look fantastic, but, at the same time, it is impossible to deny their validity. All this makes the work by Zhuomin Zhang et al. of great interest to a broad community. Therefore, the revised version of the paper could be published in Nature Communications in its current form.

Reviewer #3 (Remarks to the Author):

The authors did a great job on revising the manuscript and address all comments and questions. I don't have any further questions to the current version.

Reviewer Comments (and change made in accordance)

Reviewer #1 (Remarks to the Author):

In this revised version authors have considered referee comments in very high amount of details and addressed all major points of concerns and recommendations satisfactorily. Minor issue remains on the nature of crystallisation mechanism, which authors have argued to be of homogenous nature and minimal role of the solid substrate. I found their arguments convincing. I recommend publication of the revised manuscript with one additional comment: English corrections especially as regards to article use. There are numerous occasions of misuse (and often no use) of the definite article 'the'. Perhaps this could be further checked at the copy editor. Overall it is an excellent paper and a joy to read.

Response: We sincerely thank the reviewer for the recommendation of our paper to be published in Nature Communications. We also appreciate the reviewer for careful reading of the manuscript and the specific comments on English corrections. Following the reviewer's suggestions, we have carefully reviewed the whole manuscript and corrected the grammar issues, especially the use of the definite article 'the'.

Reviewer #2 (Remarks to the Author):

Zhuomin Zhang and co-authors have made an enormous piece of work providing exhaustive explanations and additional data verifying and validating their impressive conclusions. New experiments successfully support the initial data. Although the suggested mechanism of the crystal growth at the nanoconfinement and under an electric field is still debatable and requires additional studies, the experimental methods and analyses are correct and do not raise doubts anymore. Indeed, the presented results look fantastic, but, at the same time, it is impossible to deny their validity. All this makes the work by Zhuomin Zhang et al. of great interest to a broad community. Therefore, the revised version of the paper could be published in Nature Communications in its current form.

Response: We are pleased that the reviewer acknowledged the novelty and significance of our research. Thank you for recommending our revised manuscript to be published in Nature Communications.

Reviewer #3 (Remarks to the Author):

The authors did a great job on revising the manuscript and address all comments and questions. I don't have any further questions to the current version.

Response: We are grateful for the reviewer's affirmation on our revised manuscript.